# Immune-mediated ECM depletion improves tumour perfusion and payload delivery

Yen Ling Yeow[1], Venkata Ramana Kotamraju[2], Xiao Wang[1], Meenu Chopra[1], Nasibah Azme[1], Jiansha Wu[1], Tobias D Schoep[3], Derek S Delaney[1] (iD), Kirk Feindel[4], Ji Li[1], Kelsey M Kennedy[5], Wes M Allen[1,5], Brendan F Kennedy[1,5], Irma Larma[4], David D Sampson[4,5], Lisa M Mahakian[6], Brett Z Fite[6], Hua Zhang[6], Tomas Friman[2], Aman P Mann[2], Farah A Aziz[7], M Priyanthi Kumarasinghe[8], Mikael Johansson[7], Hooi C Ee[7], George Yeoh[1], Lingjun Mou[7], Katherine W Ferrara[6], Hector Billiran[2,†], Ruth Ganss[1], Erkki Ruoslahti[2] & Juliana Hamzah[1,*] (iD)

## Abstract

High extracellular matrix (ECM) content in solid cancers impairs tumour perfusion and thus access of imaging and therapeutic agents. We have devised a new approach to degrade tumour ECM, which improves uptake of circulating compounds. We target the immune-modulating cytokine, tumour necrosis factor alpha (TNFα), to tumours using a newly discovered peptide ligand referred to as CSG. This peptide binds to laminin–nidogen complexes in the ECM of mouse and human carcinomas with little or no peptide detected in normal tissues, and it selectively delivers a recombinant TNFα-CSG fusion protein to tumour ECM in tumour-bearing mice. Intravenously injected TNFα-CSG triggered robust immune cell infiltration in mouse tumours, particularly in the ECM-rich zones. The immune cell influx was accompanied by extensive ECM degradation, reduction in tumour stiffness, dilation of tumour blood vessels, improved perfusion and greater intratumoral uptake of the contrast agents gadoteridol and iron oxide nanoparticles. Suppressed tumour growth and prolonged survival of tumour-bearing mice were observed. These effects were attainable without the usually severe toxic side effects of TNFα.

**Keywords** extracellular matrix; immune cells; peptide; solid tumour; tumour necrosis factor alpha
**Subject Categories** Cancer; Pharmacology & Drug Discovery; Vascular Biology & Angiogenesis

## Introduction

A dense network of highly disorganised ECM is the central feature of desmoplastic tumours and often encountered in aggressive and treatment-resistant cancers (Pickup *et al*, 2014). Tumour ECM is made of overproduced scaffolds of collagen, non-collagenous glycoproteins and glycosaminoglycans (Mouw *et al*, 2014). Collectively, these ECM components represent an obstructive physical barrier that determines how migrating cells and circulating compounds enter and exit a tumour. In diagnostic and therapeutic settings, the ECM barrier restricts penetration, and thereby tumour uptake of various imaging agents and anti-cancer therapeutics (Hellebust & Richards-Kortum, 2012; Salmon *et al*, 2012; Choi *et al*, 2013). Destroying the fibrotic ECM barrier to increase the vulnerability of a tumour presents a compelling adjuvant strategy. For instance, ECM can be disrupted by employing locally injected or ectopically expressed enzymes (Guedan *et al*, 2010; Caruana *et al*, 2015), systemic application of modified hyaluronidase (PEGPH20) (Provenzano *et al*, 2012), fibrinolytic therapy with tissue plasminogen activator (Kirtane *et al*, 2017), treatment with inhibitors of ECM production (Olive *et al*, 2009; Diop-Frimpong *et al*, 2011; Vennin *et al*, 2017) or physical means, such as photothermal therapy (Marangon *et al*, 2017).

Thus far, however, existing approaches lack the specificity to target and degrade tumour ECM and, therefore, tend to cause systemic toxicity (Kirtane *et al*, 2017). Moreover, some of the strategies used to degrade or reduce tumour ECM have also been implicated in promoting metastases (Ozdemir *et al*, 2014; Sevenich & Joyce, 2014; Schmaus & Sleeman, 2015; Rath *et al*, 2017). Thus,

1 Harry Perkins Institute of Medical Research, Centre for Medical Research, QEII Medical Centre, The University of Western Australia, Perth, WA, Australia
2 Cancer Research Center, Sanford Burnham Prebys Medical Discovery Institute, La Jolla, CA, USA
3 Telethon Kids Institute, Subiaco, WA, Australia
4 Centre for Microscopy, Characterisation & Analysis, The University of Western Australia, Perth, WA, Australia
5 Department of Electrical, Electronic & Computer Engineering, School of Engineering, The University of Western Australia, Perth, WA, Australia
6 Department of Biomedical Engineering, University of California Davis, Davis, CA, USA
7 Sir Charles Gairdner Hospital, Perth, WA, Australia
8 PathWest Laboratory Medicine, QE2 Medical Centre, Perth, WA, Australia
   *Corresponding author. Tel: +61 8  6151 0732; E-mail: juliana.hamzah@perkins.uwa.edu.au
   †Present address: Department of Biology, Xavier University of Louisiana, New Orleans, LA, USA

new approaches to dealing with the ECM barrier in tumours are needed. We have devised a strategy that is based on targeting of an immune-modulating cytokine into tumour ECM. Screening of phage-displayed peptide libraries in live mice has been used to identify peptides that specifically recognise tumours (Ruoslahti, 2016). The tumour-homing peptides identified in this manner can be used in selective delivery of payloads to tumours. For example, targeted delivery of cytokines into tumours can improve therapeutic outcomes (Hamzah et al, 2008; Johansson et al, 2012; Johansson-Percival et al, 2015). Here, we identified a peptide that specifically recognises tumour ECM and used it to target a cytokine to the ECM.

TNFα is a pleiotropic cytokine known to promote innate and adaptive immune responses (Talmadge et al, 1987). It can stimulate multiple types of immune cells to secrete proteases that are capable of degrading ECM (Vaday et al, 2000; Melamed et al, 2006). An in vitro study shows that TNFα bound to fibronectin in ECM attracts monocytes and triggers their activation into MMP9-secreting cells (Vaday et al, 2000). We hypothesised that TNFα, when specifically targeted to tumour ECM, may have similar effects, leading to ECM degradation and enhanced penetration of compounds to tumours. Here, we demonstrate that TNFα fused to a new tumour ECM-recognising peptide specifically accumulates in desmoplastic tumours, increasing immune cell accumulation and ECM degradation.

## Results

### Identification and analysis of a tumour ECM-binding peptide

We designed a phage library screening protocol aimed at identifying peptides specific for tumour ECM. The screen consisted of an initial in vitro biopanning of a library of random seven-amino acid peptides flanked by a cysteine residue on each side (general structure: CX7C) on Matrigel™. Matrigel is an ECM preparation derived from a mouse tumour that produces copious amounts of basement membrane (BM)-type ECM consisting primarily of laminin, nidogen-1 (also known as entactin) and collagen IV. There are also traces of heparan sulphate proteoglycan (perlecan), along with some growth factors. The enriched phage pool from 3 in vitro rounds was subsequently subjected to 4 rounds of in vivo screening in mice bearing MDA-MB-435 human breast cancer xenograft tumours. A 9-amino acid peptide, CSGRRSSKC (termed CSG), and its variants were

present in multiple copies in the final phage pool (Appendix Fig S1A–D). CSG was selected for further study.

We compared the binding of synthetic carboxyfluorescein (FAM)-labelled CSG to tumour sections. Appendix Fig S1E and F shows robust binding to sections of neuroendocrine pancreatic tumours from genetically engineered RIP1-Tag5 mice which are strongly fibrotic (Ganss & Hanahan, 1998). CREKA, a previously identified peptide that binds to fibrin deposited on the vessel walls of tumour vessels and to tumour stroma (Simberg et al, 2007), showed weaker (about 10-fold less) binding which was essentially restricted to tumour vessels, and there was no detectable binding of a peptide with no tumour-binding activity (ARA). When injected intravenously (i.v.) into tumour-bearing mice, CSG specifically accumulated in orthotopically implanted murine 4T1 breast and RIP1-Tag5 tumours (Fig 1). Excluding the kidneys where circulating peptides are excreted, normal organs showed only background fluorescence (Fig 1A). The specificity of the tumour accumulation was confirmed by histological detection and quantification of FAM-peptide homing using an antibody against fluorescein (Fig 1B and C). The accumulation of CSG in these tumours was at least 10- to 15-fold higher than in the normal tissues. CSG also accumulated in tumours in other mouse models, including the ALB-Tag hepatocellular carcinoma (HCC) (Ryschich et al, 2006), MMTV-PyMT breast carcinoma and transplanted CT26 colon carcinoma (Appendix Fig S1G).

To determine whether CSG also recognises human tumours, we assessed FAM-CSG binding on fresh biopsies of primary human breast carcinomas from seven mastectomy patients. CSG bound selectively in all tumour specimens (Figs 1D and E, and EV1A), whereas there was negligible binding to the adjacent normal breast tissue (Fig 1D). The FAM-CSG binding was inhibited by unlabelled CSG, and the ARA control peptide showed no binding to the tumours (Figs 1D and E, and EV1A). CSG-specific tumour binding was also detected on fresh biopsies of primary human pancreatic adenocarcinoma and HCC from three pancreatectomy and hepatectomy patients, respectively (Fig EV1B). Therefore, CSG binding is conserved in human fibrotic cancers.

As CSG was first isolated based on its in vitro binding to Matrigel, we used a CSG affinity matrix to isolate the CSG target molecule from a dilute solution of Matrigel. Elution of the affinity matrix with soluble CSG peptide produced several bands, which were identified by mass spectrometry as laminin subunits alpha-1 and gamma-1, and nidogen-1. These proteins were absent in eluates

**Figure 1. CSG specifically recognises mouse and human tumours, and binds to ECM.**

A–C Mice bearing orthotopically implanted 4T1 breast cancers and RIP1-Tag5 (RT5) tumours were i.v. injected with 0.1 μmol of FAM-CSG, and tissues were collected after 1-h circulation. (A) Photographic image of tissues from 4T1 tumour-bearing mouse under bright light and UV illuminator. (B and C) Distribution of FAM-CSG in different tissues including tumours (4T1 T and RT5 T), kidney (K), vertebrae (V), lung (LG), liver (LV), intestine (I), muscle (MU), spleen (SP), heart (H), pancreas (P), brain (B), lymph node (LN) and skin (SK), detected by immunoperoxidase staining with anti-FITC antibody. Representative staining (brown) is shown for each tissue in (B) and as mean ± SEM of percentage area per tissue section stained with anti-FITC antibody (n = 3; *P < 0.05 and **P < 0.005, tumour compared to other tissues except kidney by one-way ANOVA test with Tukey's correction) in (C). Scale bars: 100 μm.

D, E Human breast tumour (Hu BT) and normal breast tissues (Hu NB): 8-μm serial tissue sections were incubated for 30 min with 1 μM FAM-CSG or FAM-ARA, in the presence or absence of 1 mM unlabelled CSG peptide. CSG (brown) was detected as in panels (B and C). (D) Representative micrographs of corresponding tissues stained with anti-FITC antibody (brown) are shown for an individual patient sample. Scale bars: 150 μm. (E) Bar charts show mean ± SEM of percentage area per tissue section stained with anti-FITC antibody (N = 4 Hu NB and N = 7 Hu BT; ***P < 0.001 and ****P < 0.0001 by one-way ANOVA test with Tukey's correction).

Source data are available online for this figure.

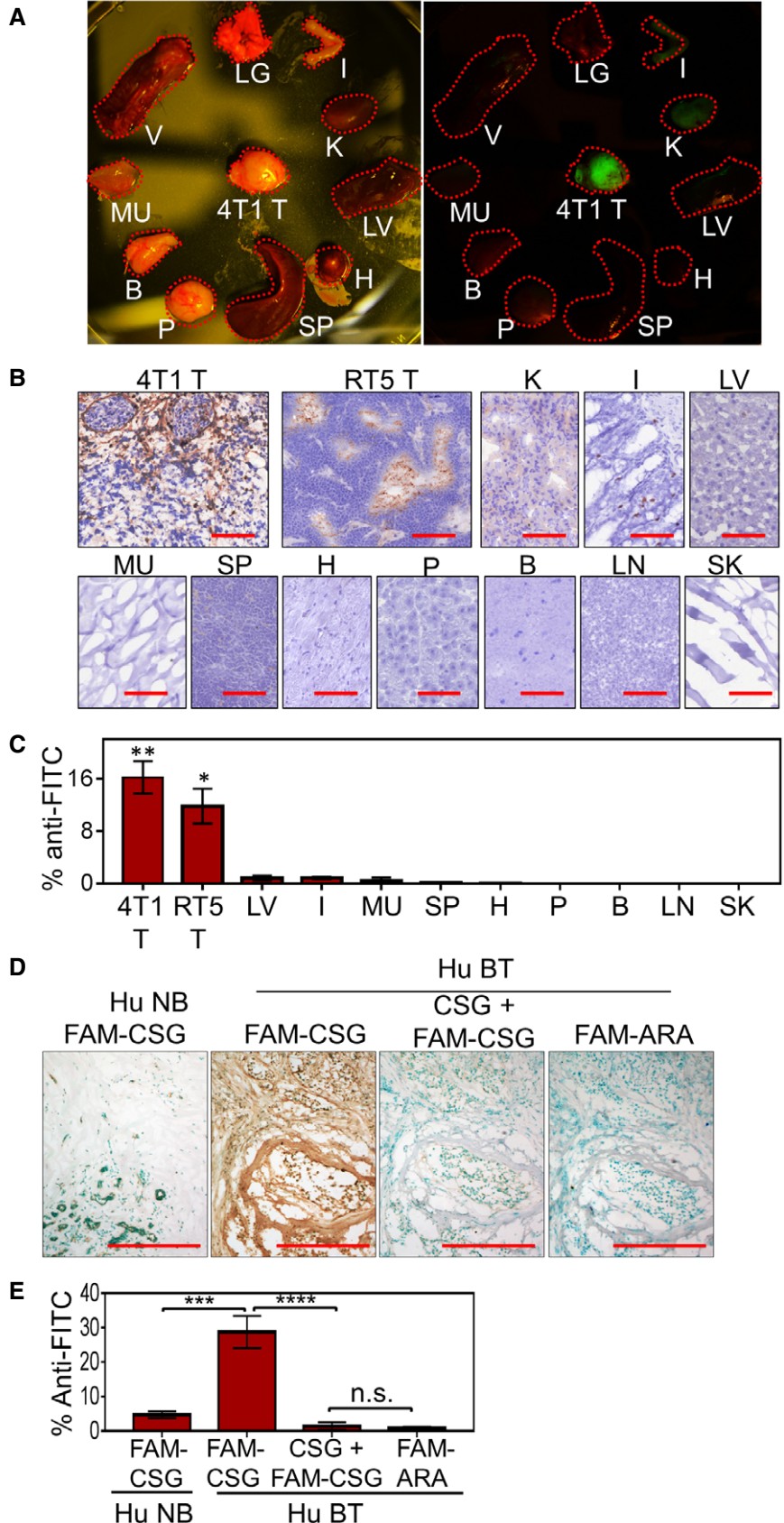

Figure 1.

obtained with the CREKA control peptide but appeared upon subsequent elution of the same matrix with CSG (Fig 2A). These results indicate that the target of CSG is laminin–nidogen-1, which exists as

a complex in ECM (Timpl *et al*, 1990). We next studied the expression of laminin and nidogen-1, as well as collagen IV, the third member of the basement membrane complex, in tumours. The three

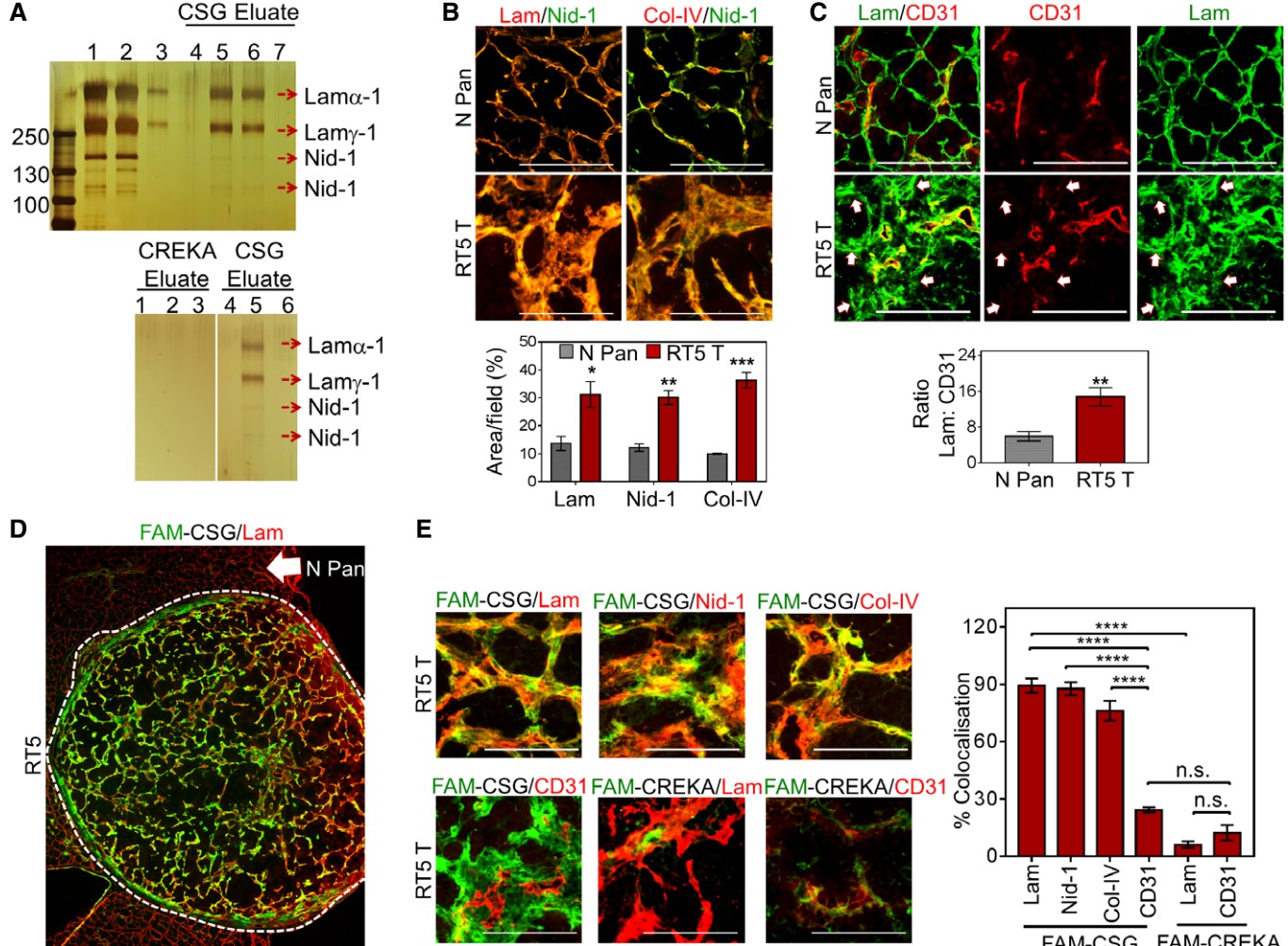

**Figure 2. CSG specifically binds tumour ECM with affinity to laminin–nidogen-1 complex.**

Matrigel extract was fractionated by affinity chromatography on CSG-coupled columns. Bound proteins were eluted with 2 mM CSG or control CREKA peptide solutions and separated by sodium dodecyl sulphate–polyacrylamide gel electrophoresis.

A   Top: Silver staining shows multiple bands in the Matrigel extract (lane 1), laminin–nidogen-1 complex (lane 2) and purified laminin (lane 3), in comparison with 4 bands eluted with CSG peptide which did not appear in the first CSG elution (lane 4) but appeared in eluted fractions 2 (lane 5) and 3 (lane 6). These bands were no longer visible in subsequent elution (lane 7). Bottom: Silver staining shows the absence of bands when CSG-coupled column was eluted with the control CREKA peptide (lanes 1–3); subsequent elution of the column with the CSG peptide shows the 4 previously shown bands (lane 5). The 4 bands (top gel, lanes 5 and 6) were identified by mass spectrometry as laminin subunit α-1 (Lamα-1), laminin subunit γ-1 (Lamγ-1) and 2 nidogen-1 bands of 140 and 110 kDa.

B   Normal pancreas from a C3H mouse (N Pan) and a RIP1-Tag5 tumour (RT5 T) were stained for the indicated ECM components. Representative micrographs are shown in the top panel. Bar graphs in the bottom panel show quantification of the area positive for each ECM protein (mean ± SEM; *n* = 3, \*$P < 0.05$, \*\*$P < 0.005$ and \*\*\*$P < 0.001$ by multiple *t*-tests).

C   Tissues as shown in (B) were stained for laminin (lam, green) and blood vessels (CD31, red). Representative micrographs are shown. The bar graph depicts the ratio of laminin over CD31 staining (mean ± SEM; *n* = 5, \*\*$P < 0.005$ by Student's *t*-test). Arrows: areas positive for laminin expression but lack CD31 staining.

D   Representative micrograph of a RIP1-Tag5 tumour (T) with normal pancreas (N Pan) showing CSG binding (green; *in vitro* binding was performed as indicated in Appendix Fig S1E), laminin staining (lam; red) and CSG–laminin co-localisation (yellow).

E   Co-staining analysis of *in vitro* bound CSG or CREKA (green) compared to indicated ECM markers or CD31+ tumour blood vessels (red). Representative micrographs (left) and corresponding bar graphs (right) show co-localisation of indicated markers with CSG or CREKA (mean ± SEM; *n* = 4, \*\*\*\*$P < 0.001$ by one-way ANOVA test with Tukey's correction).

Data information: Scale bars for (B, C and E): 50 μm. Scale bars (D): 200 μm.
Source data are available online for this figure.

proteins co-localised in all tissues; however, their abundance was greater in tumours than in normal tissues (Figs 2B and EV2A). As indicated by co-staining for laminin and the blood vessel marker CD31 (Fig 2C), laminin in normal mouse envelops the blood vessels as part of the basement membrane complex. In contrast, in tumours the basement membrane complex showed widespread expression separate from the vessels (Fig 2C) and showed strong overlaps with fibrillar collagens (collagen I and trichrome staining; Fig EV2B and C). These results show that tumour ECM differs from normal ECM in abundance and location.

FAM-CSG binding on tissue sections *in vitro* correlated with the location of laminin, nidogen-1, collagen IV and collagen I but not CD31$^+$ blood vessels in mouse and human tumours and was negligible in the basement membrane of normal tissues (Figs 2D and E, and EV2D–F). CSG binding also showed some co-localisation with ER-TR7, an antigen that recognises reticular fibroblasts and fibres, but mostly in non-cellular ECM (Fig EV2F). Consistent with the affinity pulldown results, CREKA showed only limited co-localisation with laminin (Fig 2E), indicating that CSG binding to laminin–nidogen complex in tumour sections is specific. To investigate further the source of tumour ECM recognised by CSG, we assessed CSG binding in cultured 4T1 cells and βTC-C3H tumour cells derived from RIP-Tag mice. Laminin expression and CSG binding were most pronounced in 4T1 cells (Fig EV2G). βTC-C3H tumour cells did not produce laminin and bind CSG (Fig EV2G), indicating that the ECM complexes that bound CSG in RIP1-Tag5 tumours (Fig 2D and E) were mainly stroma-associated.

### Immune-enhancing effects of TNFα-CSG

To evaluate CSG as a carrier molecule for TNFα delivery to tumour ECM, we produced bacterial recombinant TNFα with carboxy-terminal CSG peptide (molecular weight 18.9 kDa) (Appendix Fig S2A). The fusion protein was biologically active as shown by induction of VCAM-1 expression in cultured brain endothelial cells (Appendix Fig S2B). Fluorescein-labelled TNFα-CSG, unlike TNFα, selectively homed to tumour matrix *in vivo* (Appendix Fig S2C), in the same manner as free CSG peptide (Fig 1).

TNFα-CSG was used in this study at a therapeutic dose range of 2–10 μg. Even at the highest dose (10 μg i.v. injection/day for 5 consecutive days), it was well tolerated in mice and did not reduce animal body weights (Appendix Fig S2D). In contrast, the untargeted TNFα is lethally toxic at 10 μg (Cauwels *et al*, 1995; Marino *et al*, 1997). Quantification of plasma C-reactive protein (CRP), a marker for acute and chronic inflammation, showed that a single dose of 10 μg untargeted TNFα significantly elevated plasma CRP in healthy animals, whereas TNFα-CSG, given at the same dose once or five times on consecutive days, did not significantly raise plasma CRP levels (Appendix Fig S2E).

To evaluate the effects of TNFα-CSG on tumours, we focused on the RIP1-Tag5 and 4T1 models because these tumours have different levels of immune infiltrates but share abundant ECM and thus CSG targets (Fig 1). In RIP1-Tag5 tumours, which were devoid of immune infiltrates, treatment with TNFα-CSG (5 daily i.v. injections of 2 or 5 μg) caused a significant influx of CD8$^+$ and CD4$^+$ T cells and macrophages (Fig 3A). Moreover, CD8$^+$ T cells selectively accumulated in ECM-rich zones in the TNFα-CSG-treated tumours (Fig EV3A). In contrast, our previously described TNFα fused to the vessel-targeting peptide RGR (Hamzah *et al*, 2008; Johansson *et al*, 2012) at similar total doses caused immune cell infiltration around tumour blood vessels (Fig EV3B).

In addition, in 4T1 tumours with pre-existing immune infiltrates, we compared the effect of TNFα and TNFα-CSG on immune cell infiltration at the maximally tolerated dose for TNFα (5 daily injections of 0.5 μg). At 0.5 μg, TNFα-CSG led to a significant increase in intratumoral CD8$^+$ T cells and macrophages (CD11b$^+$ F480$^+$ Ly6G$^-$) but reduced the population of tumour myeloid-derived suppressor cells (CD11b$^+$ F480$^-$ Ly6G$^+$) (Fig EV3C). TNFα had no effect on intratumoral T cells or myeloid-derived suppressor cells, although an increase in macrophage level was also observed (Fig EV3C). In addition to RIP1-Tag5 and 4T1 tumours, TNFα-CSG (5 daily i.v. injections of 5 μg) also improved immune T-cell infiltration into ALB-Tag HCC (Fig EV3D). Therefore, ECM targeting of TNFα, in contrast to unconjugated TNFα, attracts T cells into the tumour microenvironment irrespective of tissue phenotypes, without inducing systemic toxicity.

### Overexpression of proteases, and reduced ECM content and stiffness in response to TNFα-CSG

Having shown that TNFα-CSG attracts immune cells to tumour ECM, we explored their potential role in tumour matrix degradation. Since previous *in vitro* studies showed that TNFα stimulates monocytes to secrete MMP9 (Vaday *et al*, 2000), we first profiled protease expression by immune cells in response to TNFα-CSG treatment. Immune cells from 4T1 tumours of mice treated with TNFα-CSG or CSG were separated by FACS into CD45$^+$/CD11b$^-$ and CD45$^+$/CD11b$^+$ cells, and expression of various proteases known to degrade ECM (Sorokin, 2010; Sevenich & Joyce, 2014) was analysed. Tumour CD45$^+$/CD11b$^-$ cells from mice treated with TNFα-CSG expressed twofold to 20-fold higher levels of mRNAs for multiple proteases than cells from the CSG-treated tumours. In particular, the mRNA transcripts including uPA, MMP2, MMP3, MMP9, MMP12, MMP14 and cathepsin L were significantly increased at 10 μg of TNFα-CSG (Fig 3B). TNFα-CSG treatment did not affect protease gene expression in tumour CD11b$^+$ cells or in cultured 4T1 tumour cells (with the exception of MMP3) (Fig EV4A). Protein-level analysis of MMP9 and MMP12 release by CD4$^+$ T cells and cathepsin L by CD8$^+$ T cells showed that TNFα-CSG specifically induced release of these proteases (Fig 3C). CD11b$^+$ cells inherently secreted high levels of proteases, especially MMP9 and MMP12 (Fig 3C). Thus, the accumulation of these cells (Figs 3A and EV3) in response to TNFα-CSG may also contribute to increased expression of proteases in the tumours.

Immunostaining analysis of ECM proteins in TNFα-CSG-treated RIP1-Tag5 tumours showed significantly reduced levels of laminin, nidogen-1 and collagen IV (Fig 4A and B), whereas treatment with the CSG peptide, or with the blood vessel-targeted TNFα-RGR, (Johansson *et al*, 2012) had no effect (Fig 4A and B). TNFα-CSG treatment also reduced other tumour ECM components, which included fibrillar collagens (stained with picrosirius red), collagen I, perlecan (HSPG2) and fibronectin (Fig EV4B). The number of stromal content stained with ER-TR7 antibody was also significantly reduced (Fig EV4B). T-cell infiltration is only observed in tumour and not in the surrounding tissue (Fig EV5A). Thus, ECM content in

normal exocrine tissue surrounding the tumours was unaffected by TNFα-CSG treatment (shown for laminin in Fig EV5A). Interestingly, the reduction in ECM content in response to TNFα-CSG treatment was positively correlated with immune cell infiltration in the RIP1-Tag5 tumours (Fig 4C and D), supporting the notion that the immune cell proteases degrade the ECM.

Tissue stiffness is a hallmark of solid tumours and a direct function of ECM content/properties (Pickup *et al*, 2014). Increased stiffness reflects the pathological state of malignant tissue that influences cell behaviour in tumour (Lampi & Reinhart-King, 2018). Quantitative micro-elastography was employed to measure structural heterogeneity of tumour stiffness (Kennedy *et al*,

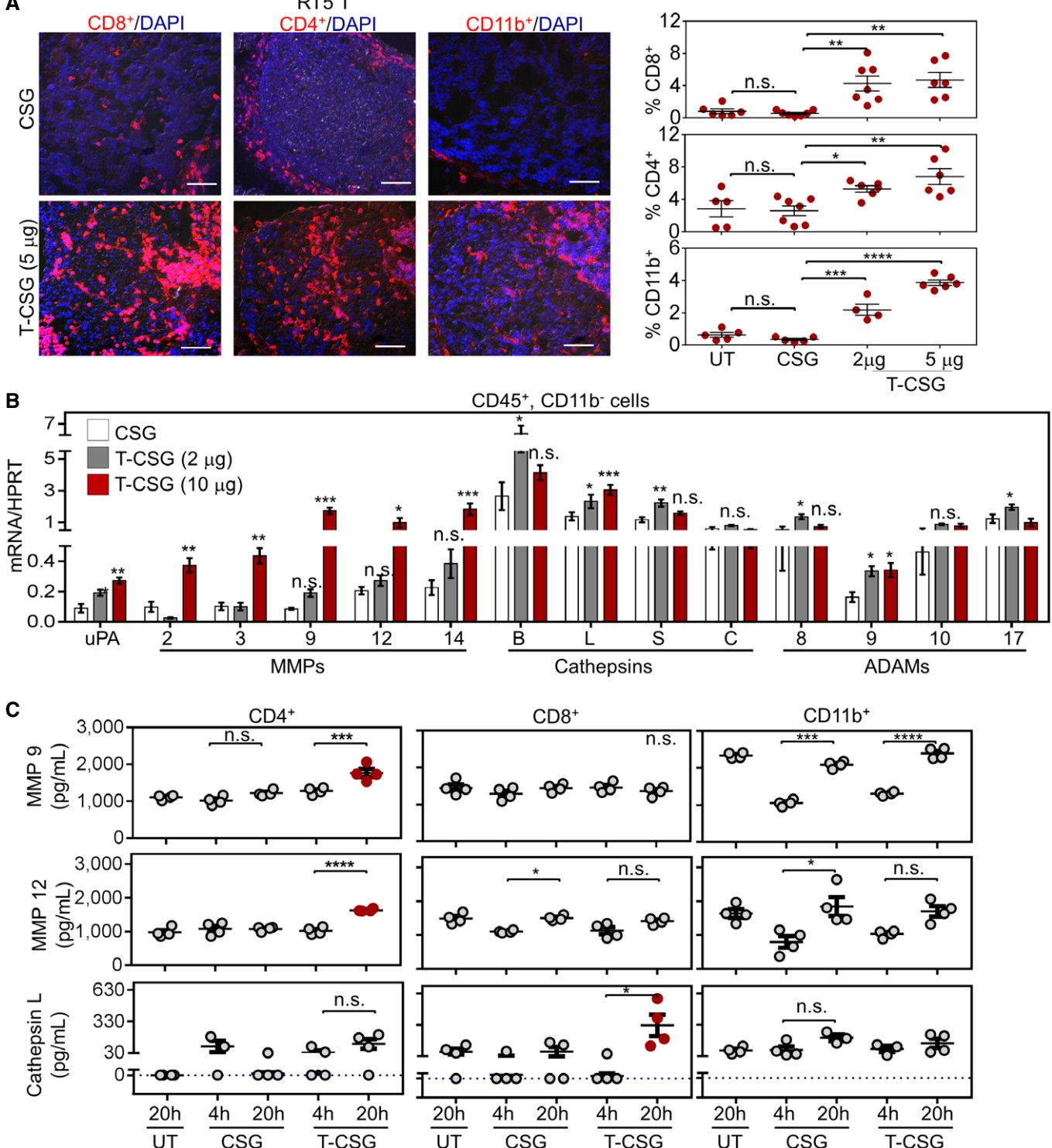

Figure 3.

**Figure 3.  TNFα-CSG (T-CSG) treatment promotes immune cell infiltration and induces them to secrete proteases.**

A  Left panels: Representative micrographs of immune cell infiltrates (red) in tumours from RIP1-Tag5 mice untreated (UT) or treated with 5 daily i.v. injections of indicated compounds. Scale bars: 100 μm. Right panels: fraction of cells stained with the indicated antibodies/tumour and mean ± SEM (4–7 tumours from $n = 3$ mice; *$P < 0.05$, **$P < 0.01$, ***$P < 0.001$ and ****$P < 0.0001$ by one-way ANOVA test with Tukey's correction).

B  Quantitative RT–PCR analysis comparing protease expression in isolated CD45+ CD11b− cells from 4T1 tumours of mice treated with 5 daily i.v. injections of indicated compounds, relative to the expression of hypoxanthine-guanine phosphoribosyltransferase (HPRT) standard. Bar graphs showing means ± SEM of the mRNA expression for the indicated proteases ($n = 4$–5; *$P < 0.05$, **$P < 0.005$ and ***$P < 0.001$ when compared to CSG-treated group by multiple $t$-tests).

C  Quantitative analysis of proteases in media of splenic T cells (CD4+ and CD8+) and CD11b+ cells isolated from C3H mice, after 4- and 20-h incubation with 0.1 μg CSG or 1 μg TNFα-CSG. A 20-h incubation of untreated (UT) cells was also included as control. Each dot represents an individual sample, and mean ± SEM is shown ($n = 4$; *$P < 0.05$, ***$P < 0.001$ and ****$P < 0.0001$ by one-way ANOVA test with Tukey's correction or Kruskal–Wallis test with Dunn's correction for cathepsin L). Horizontal dotted line indicates below the limit of detection.

Source data are available online for this figure.

2015a). The mean tumour stiffness remains unchanged in response to TNFα-CSG treatment (Fig 5; Appendix Table S1). However, control tumours showed greater structural heterogeneity of tissue stiffness displaying more frequent "hotspots" of stiffest areas (exceeded 100 kPa) and also some areas in the softest category (under 10 kPa). TNFα-CSG treatment resulted in a more homogenous distribution of tumour stiffness, with significantly reduced stiffness variability in the range well below 100 kPa (Fig 5A–D). Stiffness measurements also demonstrated reduced tumour stiffness correlated with TNFα-CSG-induced ECM loss (Figs 5E and EV5B).

**Reduced ECM content and tumour stiffness correlate with improved tumour perfusion and uptake of nanoparticle contrast agents**

Lowering ECM content is known to lead to improved blood perfusion and drug delivery (Provenzano et al, 2012; Jain et al, 2014; Kirtane et al, 2017). Indeed, blood vessels of TNFα-CSG-treated tumours when labelled by i.v. injection of FITC–lectin were significantly dilated compared to vessels in control CSG-treated or TNFα-RGR-treated tumours (Figs 6A and EV5C). In addition, whilst the treatment has no effect on CD31+ vessel numbers (Fig EV5D), the overlay of lectin+ vessels with a CD31 stain, a measure of tumour perfusion, was significantly enhanced in TNFα-CSG treatment groups (Figs 6B and EV5E). There was an inverse correlation between lectin staining and tumour collagen IV content (Fig 6C), which is in agreement with earlier studies showing that ECM degradation enhances tumour vessel perfusion, presumably by decompression (Provenzano et al, 2012; Kirtane et al, 2017).

To assess the impact of reduced tumour ECM on tumour perfusion and the uptake of circulating compounds into tumours, we measured transport of the gadolinium contrast agent gadoteridol from the circulation into 4T1 tumours by dynamic contrast-enhanced MRI (DCE-MRI) in real time in vivo. Gadolinium uptake rate doubled in the TNFα-CSG-treated tumours as compared with the CSG control (Fig 6D). Uptake of iron oxide nanoparticles (IONPs) also increased in response to TNFα-CSG therapy, as shown by T2* and T2 relaxation images (Figs 6E and EV5F). Increased accumulation of IONPs in TNFα-CSG-treated tumours was also confirmed by immunohistochemistry (Fig 6F). Collectively, these data show that reduced ECM content and cancer stiffness are associated with enhanced tumour uptake of compounds ranging from small molecules such as gadoteridol to nanoparticles.

**TNFα-CSG enhances anti-tumour immune responses without increasing metastasis**

To study the potential therapeutic effects of TNFα-CSG, we first assessed the survival of RIP1-Tag5 and 4T1 tumour-bearing mice. A treatment regimen with 5 daily doses of 2–5 μg TNFα-CSG prolonged the survival of these mice (Fig 7A and B), consistent with reduced 4T1 tumour burden (Fig 7C) and cancer cell proliferation (Ki67+ staining; Appendix Fig S3A), and increased apoptosis (TUNEL+ cells; Appendix Fig S3B). Direct exposure of cultured 4T1 and RIP1-Tag5 tumour cells to TNFα-CSG, even at concentrations sevenfold to 18-fold higher than estimated in vivo levels, did not significantly reduce cell viability (Appendix Fig S3C). Thus, the in vivo therapeutic data, along with evidence of high infiltration of T cells in TNFα-CSG-treated tumours (Figs 3A and EV3C), suggest the potential involvement of T cells in mediating anti-tumour immunity.

This prompted us to quantify cytotoxic (granzyme B+) T cells in TNFα-CSG-treated 4T1 tumours. Indeed, TNFα-CSG therapy increased the abundance of cytotoxic T cells but not immune-suppressing regulatory (CD4+ CD25+ FOXP3+) T cells in the syngeneic 4T1 tumour model (Fig 7D; gating strategies in Appendix Fig S3D). To confirm the involvement of T cells in anti-tumour immunity, we treated immunodeficient 4T1 tumour-bearing BALB/c nude mice with TNFα-CSG in the absence or presence of adoptively transferred naive splenic T cells (Fig 8A). In the absence of T cells or in the presence of only CD4+ or CD8+ T cells, TNFα-CSG treatment did not significantly reduce tumour weight and volume (Fig 8B). However, TNFα-CSG treatment and adoptive transfer of both CD8+ and CD4+ T cells (triple combination) produced anti-tumour effects similar to those observed in syngeneic 4T1 tumour-bearing mice (Fig 8B). Similarly, tumours only showed significant infiltration of T cells, ECM loss and inhibition of cell proliferation under triple treatment (Fig 8C and D, and Appendix Fig S3E). These data suggest both CD8+ and CD4+ T cells are required for intratumoral TNFα-CSG effects.

An important consideration for any ECM-reducing strategy is the potential risk of metastatic dissemination (Ozdemir et al, 2014; Sevenich & Joyce, 2014; Schmaus & Sleeman, 2015; Rath et al, 2017). Appendix Fig S4A shows that TNFα-CSG-treated 4T1 tumours were less hypoxic especially in areas containing wider blood vessels, which reduces the likelihood of hypoxia-driven metastatic dissemination (Rankin & Giaccia, 2016). We further assessed the effect of TNFα-CSG on highly metastatic 4T1 tumour cells (Appendix Fig S4B) which typically colonise in the lung from as early as 19 days after orthotopic implantation in the mammary fat

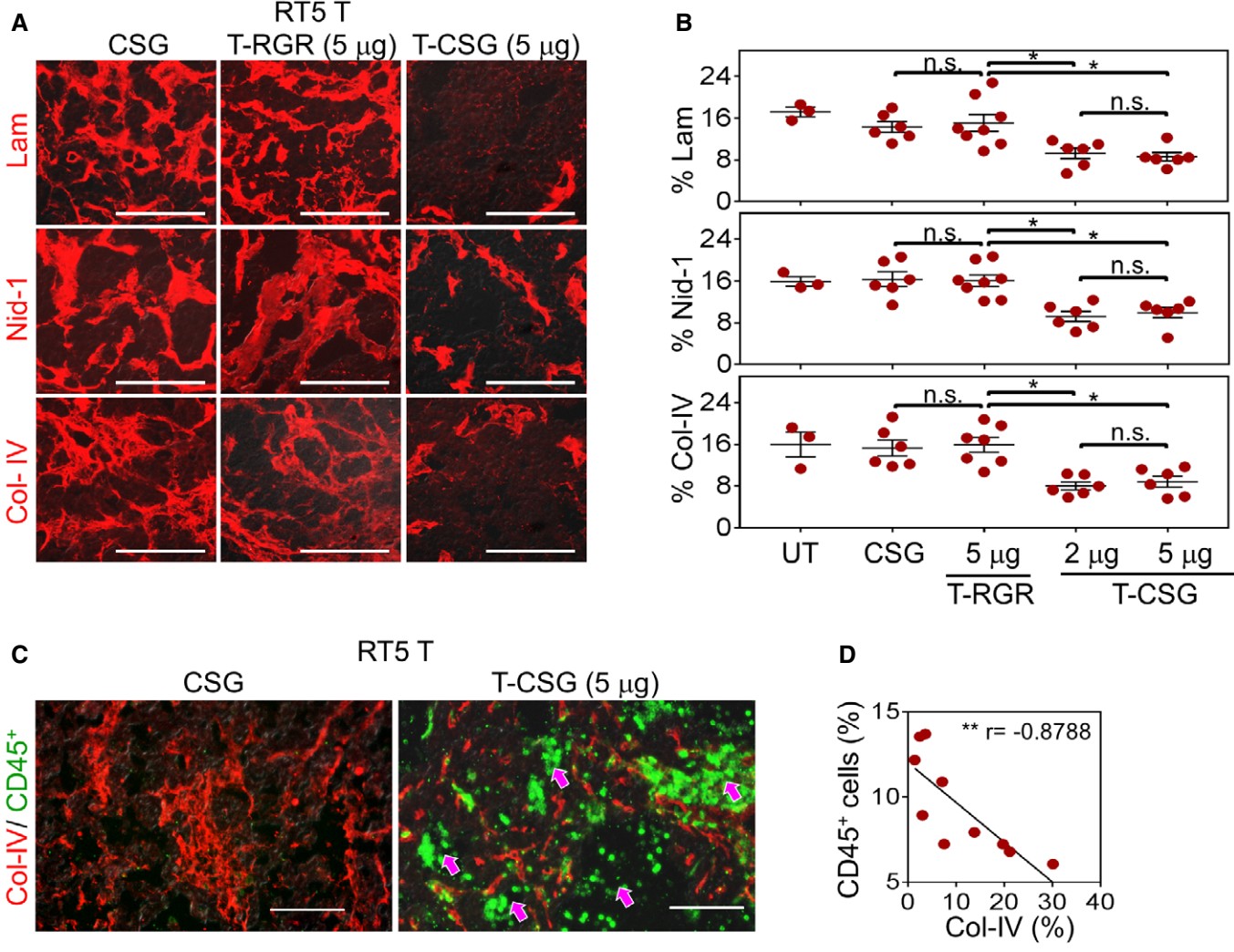

**Figure 4. TNFα-CSG treatment induces specific tumour ECM degradation.**

Tumour sections from RIP1-Tag5 following treatment with 5 daily i.v. injections of indicated compounds were analysed for ECM expression.

A   Staining with indicated antibodies (red) for tissues treated with 5 μg doses of TNFα-CSG (T-CSG) or TNFα-RGR (T-RGR), or 0.8 μg doses of CSG peptide. Scale bars: 50 μm.

B   Quantitative analysis of ECM staining/field/tumour from panel (A) and mean ± SEM (3–8 tumours from $n = 3$ mice; *$P < 0.05$ for 2 and 5 μg TNFα-CSG compared to other groups by one-way ANOVA test with Tukey's correction).

C   Co-staining analysis of tumours (as in panel A) with collagen IV (ECM, red) and CD45 (immune infiltrate, green). Arrows: areas within tumour showing immune cells in sites negative for collagen IV expression. Scale bars: 50 μm.

D   Correlation analysis of % CD45[+] cells and collagen IV[+] areas/tumour from TNFα-CSG-treated RIP1-Tag5 mice (10 tumours from $n = 3$ mice; Spearman's correlation, $r = -0.8788$, **$P = 0.0016$).

Source data are available online for this figure.

pad (Pulaski & Ostrand-Rosenberg, 2001). We treated mice bearing 4T1 tumours with a TNFα-CSG regimen that resulted in high expression of proteases (Fig 3B) and measured lung metastasis by colony assays (Pulaski & Ostrand-Rosenberg, 2001). The treatment significantly reduced dissemination of 4T1 cells from primary tumours to the lungs (Appendix Fig S4C).

In summary, our findings show that treatment of tumour-bearing mice with TNFα targeted to tumour laminin–nidogen complexes is well tolerated and induces tumour infiltration of immune cells, which in turn results in loss of tumour ECM, improved tumour perfusion, reduced tumour burden and enhanced overall survival.

# Discussion

We have constructed a recombinant TNFα fusion protein that specifically localises to tumour laminin–nidogen complexes when injected systemically into mice bearing desmoplastic tumours. By eliciting an influx of immune cells into the tumours, the fusion protein causes dramatic reduction in desmoplasia. The reduced ECM content improves tumour perfusion, thus facilitating the entry of circulating compounds, such as imaging agents into the tumour. Treatment also suppresses tumour growth which is only seen in immunocompetent mice, suggesting activation of adaptive

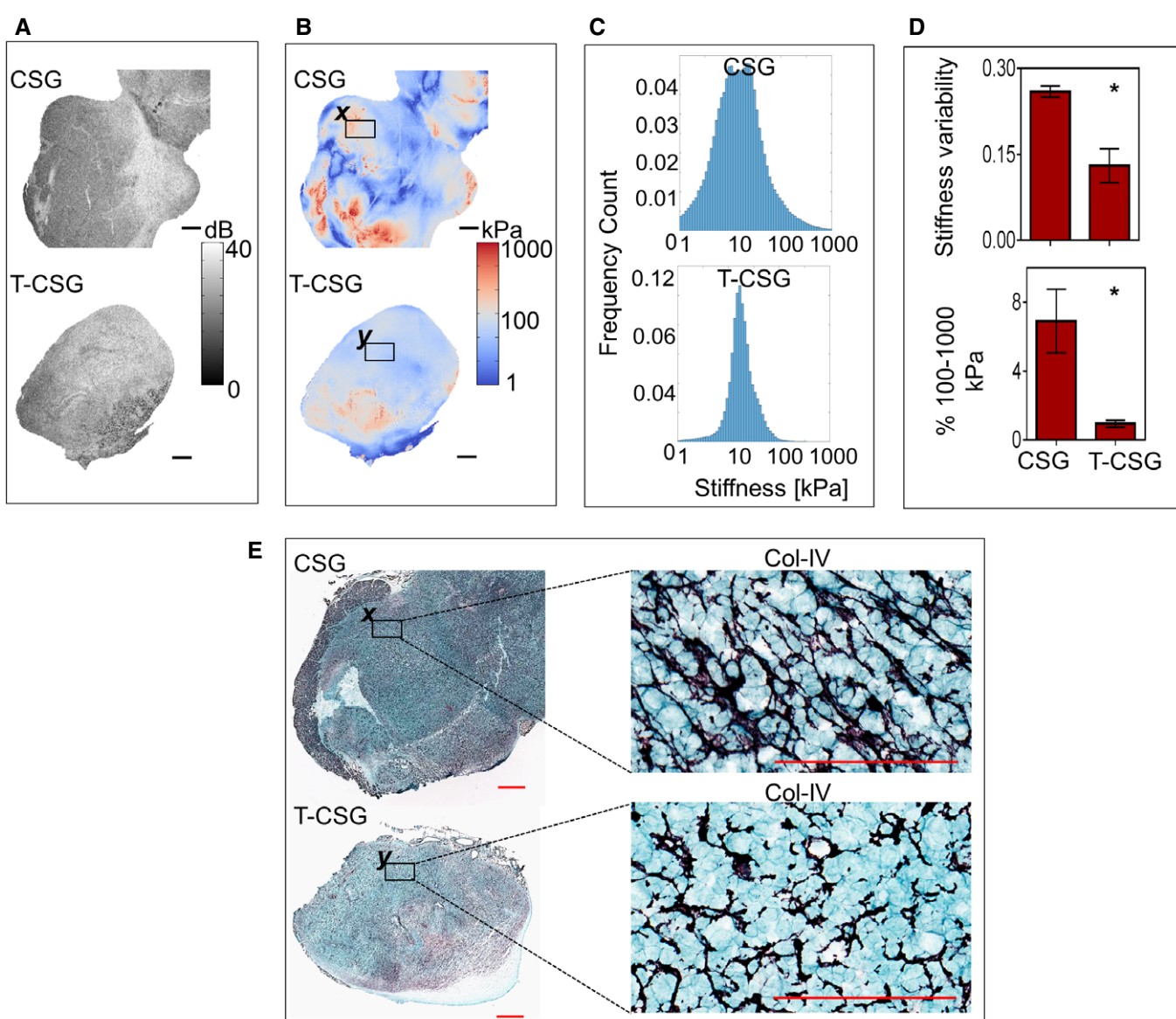

**Figure 5. TNFα-CSG treatment reduces tumour stiffness.**

Stiffness analyses of the 4T1 tumours following treatment with 4 × daily i.v. injections of indicated compounds. Tumour stiffness was analysed on day 5.

A   En face optical coherence tomography (OCT) images of tumours at a depth of ~100 μm. Scale bars: 1 mm.
B   The corresponding en face quantitative micro-elastograms showing tumour elasticity. Scale bars: 1 mm.
C   Plots of stiffness distribution in the tumours.
D   Top: Stiffness variance (see Materials and Methods). Bottom: Fractions of pixels between 100 and 1,000 kPa. For both plots, mean ± SEM is shown ($n = 3$; *$P < 0.05$ by Student's *t*-test).
E   Collagen IV staining (purple = collagen) of whole tissue sections. Scale bars: 1 mm and 100 μm. Selected regions (*x* and *y* in B and E) were compared for collagen IV staining at higher magnification (E) and the corresponding micro-elastograms (B).

Source data are available online for this figure.

tumour immunity. These effects are achieved without systemic toxicity.

The CSG peptide we used for tumour targeting has several characteristics that distinguish it from other tumour-homing peptides. Whilst most tumour-specific homing peptides (and antibodies) target cell surface proteins specifically expressed on tumour vasculature or tumour cells, we selected CSG for ECM targeting. In the few cases where ECM components have been targeted before, target proteins such as tumour-specific variants of the ECM proteins fibronectin and tenascin are primarily located around tumour blood vessels (Borsi *et al*, 2003; Kim *et al*, 2012). In contrast, CSG and its payload appear to penetrate into tumours reaching sites far from the blood vessels. In this regard, CSG resembles the previously discovered CendR-type tumour-penetrating peptides, which activate an

endocytic transcellular and transtissue transport pathway (Ruoslahti, 2016). Interestingly, the CSG sequence, CSGRRSSKC, does contain a potential CendR motif (RSSK), but it remains to be seen whether the CendR transport pathway is responsible for the tumour-penetrating properties of CSG.

The molecular structure in ECM recognised by CSG may be universal to tumours since all mouse and human tumours analysed so far showed specific and strong CSG homing or binding. The laminin–nidogen complex we identified as the target of CSG is a major constituent of all basement membranes, which also

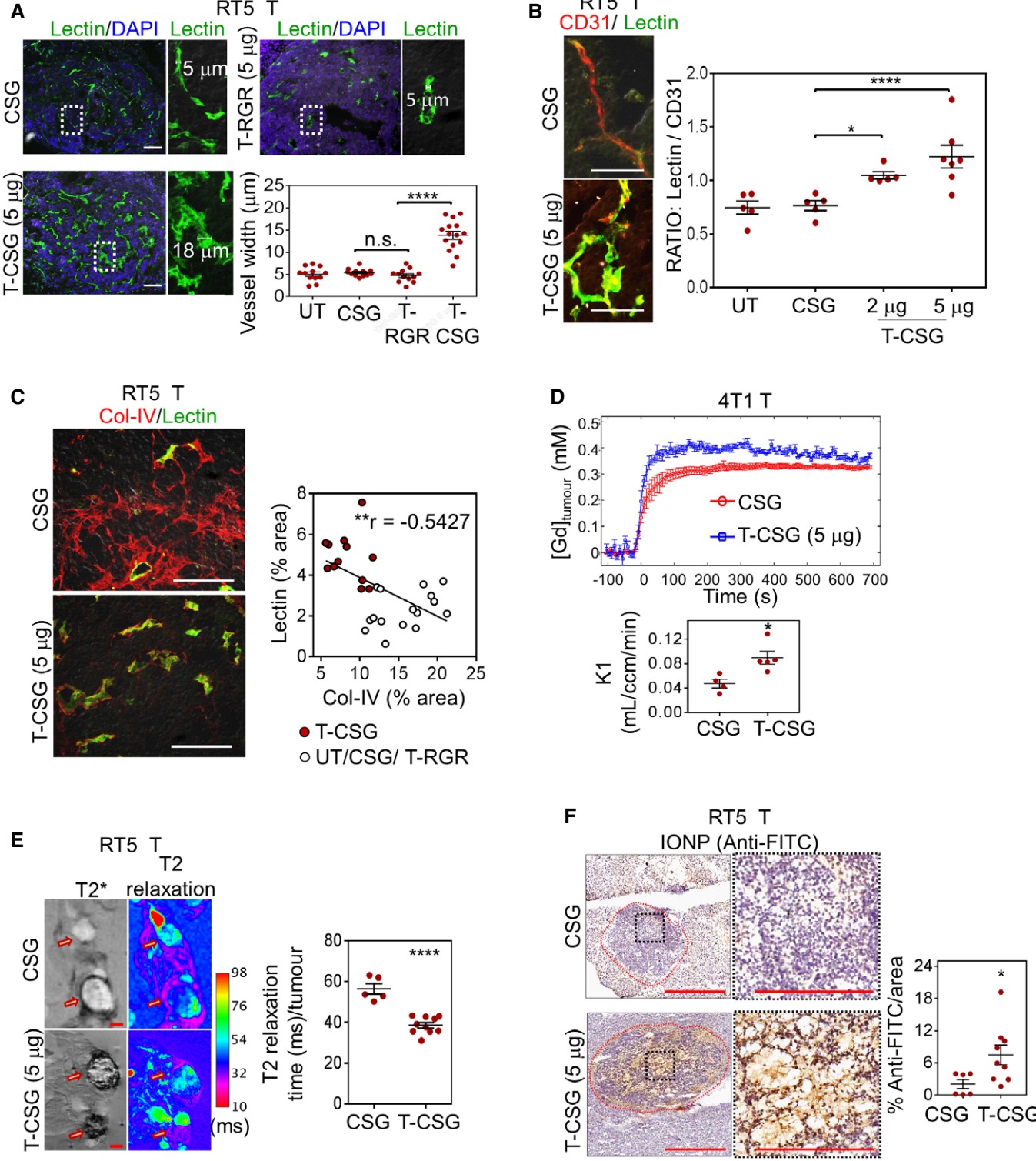

**Figure 6.**

◀

**Figure 6.  ECM depletion in TNFα-CSG-treated tumours (T-CSG) correlates with decompressed blood vessels, and enhanced perfusion and uptake of nanoparticles.**

A  Lectin[+] vessels (green) in tumours after RIP1-Tag5 mice were treated with 5 daily i.v. injections of indicated compounds (or untreated, UT) and perfused with fluorescein-labelled lectin. Representative micrographs show lectin[+] vessel widths for each group, and quantification of mean vessel diameter/tumour area and mean ± SEM are shown (12–15 tumours from $n = 5$ mice; ****$P < 0.0001$ for TNFα-CSG compared to control groups by one-way ANOVA test with Tukey's correction). Scale bars: 100 μm.

B  Co-staining of lectin-perfused RIP1-Tag5 tumours (as in panel A) with CD31 (red, blood vessels). Left: Micrographs show co-localisation (yellow) of perfused lectin and CD31 staining in tumours. Scale bars: 50 μm. Right: Data show lectin:CD31 ratio for individual tumours and mean ± SEM (5–7 tumours from $n = 3$ mice; *$P < 0.05$, ****$P < 0.0001$ by one-way ANOVA test with Tukey's correction).

C  Left: Representative tumour sections (as in panel A) show lectin[+] vessel and co-stained collagen IV (red). Scale bars: 50 μm. Right: Correlation analysis of lectin[+] and collagen IV[+] staining for all tumours irrespective of treatment (Pearson correlation, $r = -0.5427$, **$P = 0.0028$).

D  Real-time measurement of 4T1 tumour uptake of gadolinium tracer. Top: Average gadolinium concentration after a bolus i.v. injection of 0.3 μmol/g gadoteridol was measured by T1-weighted MRI and plotted against time after mice were treated with indicated compounds. Bottom: The rate constant, K1, representing permeability into the tumour (see Materials and Methods) is plotted for individual mice and mean ± SEM ($n = 4$–5; *$P < 0.05$ by Student's $t$-test).

E  Left: *Ex vivo* MRI scans of T2* (dark contrast) and T2 relaxation (magenta) show detectable amounts of IONP in tumours 4 h after an i.v. injection of 5 mg/kg IONP in RIP1-Tag5 mice treated as in panel (A) (arrow: tumour nodule). Scale bars: 1 mm. Right: Reduction in T2 relaxation time indicates the relative amount of IONP in individual tumours and mean ± SEM (5–10 tumours from $n = 3$ mice; ****$P < 0.0001$ by Student's $t$-test).

F  Left: Micrographs show detection of IONP in RIP1-Tag5 tumours based on immunostaining with anti-FITC-HRP antibody. Scale bars: 400 μm and 200 μm for higher magnification images. Right: Plots of IONP in individual tumours detected by reactivity with anti-FITC-HRP and mean ± SEM are shown (6–9 tumours from $n = 3$ mice; *$P < 0.05$ by Student's $t$-test).

Source data are available online for this figure.

include collagen IV and perlecan (Aumailley *et al*, 1993). In normal tissues, these complexes are restricted to a thin basement membrane that supports vessels and epithelia, whereas, in tumours, an extensive network is formed throughout the tumour that is not associated with blood vessels. There is a clear quantitative difference in the amount of ECM; in the cancers we studied, tumours expressed twofold to 10-fold more ECM than the corresponding normal tissue. This difference, at least at the lower end, does not entirely explain the lack of CSG homing/binding to normal tissues. Importantly, CSG binds to both stroma-associated ECM and ECM secreted by neoplastic cells. These findings raise the possibility of a structural difference in tumour ECM compared to normal ECM that allows CSG binding. As tumour ECM complexes are expressed in an aberrant manner, it could also be disorganised, exposing a normally hidden epitope. Such a situation has been described in collagen I of fibrotic lungs (Desogere *et al*, 2017). In this scenario, collagen I is overproduced and assembled into disorganised fibrils, which in turn exposes binding sites for a specific peptide probe.

Our CSG-based targeting strategy was employed to achieve high levels of TNFα locally and specifically associated with tumour ECM. In contrast, untargeted TNFα is highly toxic when administered systemically (Cauwels *et al*, 1995; Marino *et al*, 1997; Lejeune *et al*, 2006). Moreover, the biological effects of TNFα targeted to tumour ECM were different from those of TNFα targeted to tumour vessels (Johansson *et al*, 2012). Targeting TNFα to tumour ECM has some major benefits: first, the amount of ECM components is high relative to binding moieties in blood vessels or tumour cells, making the targeting particularly effective. Second, TNFα-CSG enables selective modulation of the matrix. Abundant ECM in cancer represents a physical barrier for drug delivery, and the ECM also provides a supportive environment for tumour growth (Hellebust & Richards-Kortum, 2012; Salmon *et al*, 2012; Choi *et al*, 2013). Moreover, particularly in highly desmoplastic tumours, ECM is extensively remodelled to form harder and stiffer tissue than the normal counterpart tissues (Mouw *et al*, 2014; Pickup *et al*, 2014). Increased ECM content and stiffness exerts a mechanical compressive force within the tumour

microenvironment, causing the vessels within the tumour to collapse (Jain *et al*, 2014). Therefore, ECM degradation releases the stresses and is considered an efficient strategy for tumour blood vessel decompression which in turn enhances tumour perfusion and access of circulating compounds (Provenzano *et al*, 2012; Jain *et al*, 2014; Rahbari *et al*, 2016; Kirtane *et al*, 2017).

Systemic delivery of ECM-degrading enzymes such as PEGPH20 (pegylated hyaluronidase) has been used to degrade tumour ECM components for the purpose of enhancing perfusion and access to solid tumours (Guedan *et al*, 2010; Provenzano *et al*, 2012; Caruana *et al*, 2015; Rahbari *et al*, 2016; Kirtane *et al*, 2017). The TNFα-CSG strategy differs from these approaches in that the ECM reduction following TNFα-CSG therapy is a result of immune cell infiltration and local production of a cocktail of ECM-degrading proteases by these immune cells. The ECM depletion effects by TNFα-CSG include the loss of laminin, nidogen-1, collagen IV, collagen I, fibrillar collagen, fibronectin, perlecan and potentially fibroblasts which are positive for ER-TR7. Whilst our data clearly indicate that TNFα-CSG can remove ECM components in advanced tumours, one limitation of this study is that the potential suppressive effect by TNFα-CSG on tumour fibroblasts, and their secretion of ECM components was not addressed.

We show that ECM reduction by TNFα-CSG alleviates tumour stiffness, reduces vessel compression, increases vessel dilation and may facilitate further increases in immune cell infiltration. Similar to what has been shown for other ECM-reducing strategies (Provenzano *et al*, 2012; Kirtane *et al*, 2017), TNFα-CSG enhanced tumour perfusion and uptake of MRI contrast agents, indicating potential use for cancer imaging. Currently, clinical MRI detection of cancer and differentiation from normal tissues are based on differences in vascularity, vessel permeability and extracellular diffusion space (Kinkel & Hylton, 2001; Wang, 2011). IONPs, including particles marketed as Feridex® and Resovist®, failed commercially as MRI contrast agents. Their poor performance, as indicated in our findings and others (Wang, 2011), is partly because IONPs do not penetrate into solid tumours. Here, we show that TNFα-CSG therapy improves the tumour accumulation of IONPs. Thus, TNFα-CSG treatment prior to imaging could potentially sensitise the tumours

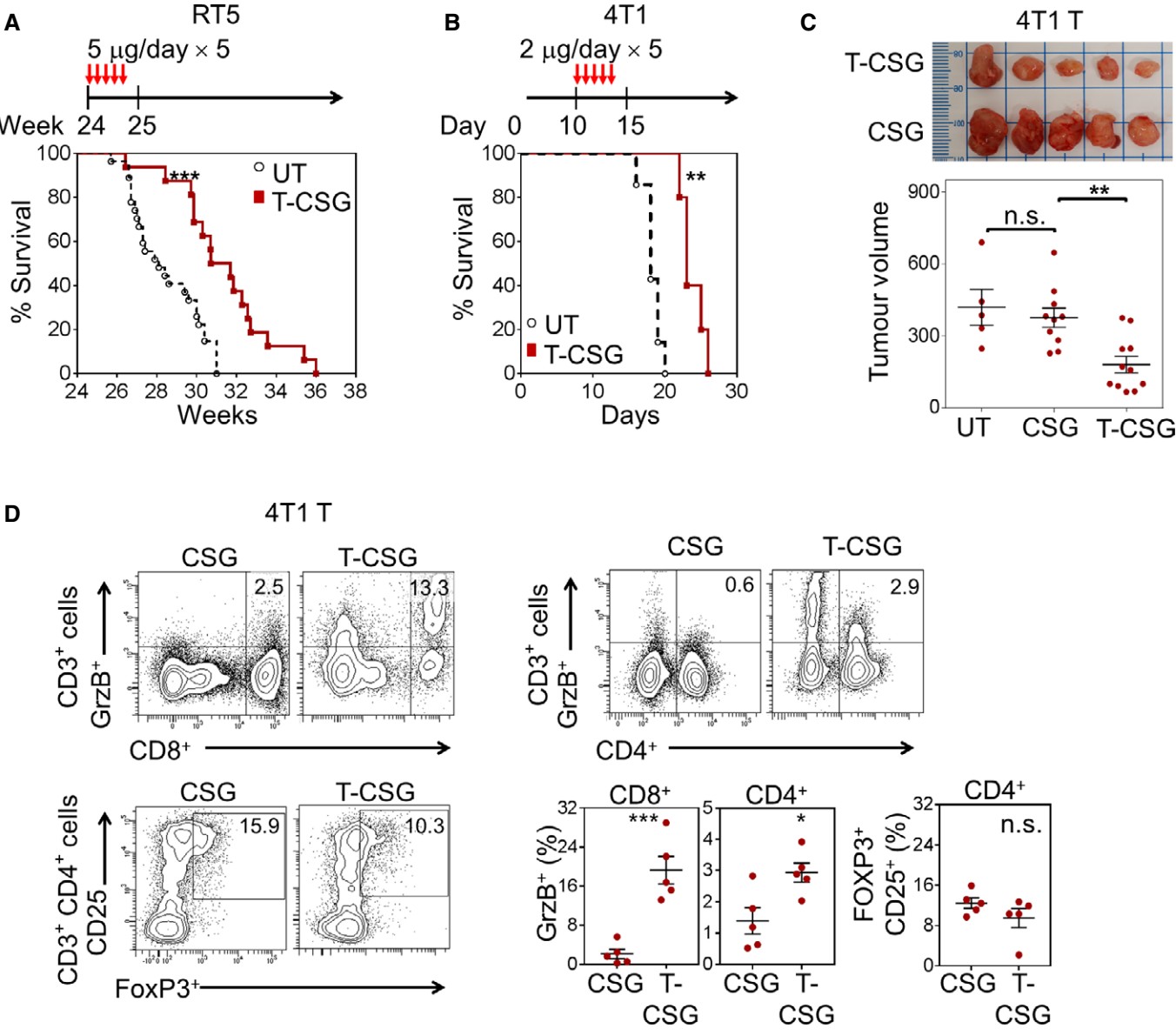

**Figure 7. TNFα-CSG therapy enhances survival, suppresses tumour growth and generates intratumoral effector T cells.**

A, B    Per cent survival of mice bearing RIP1-Tag5 and 4T1 tumours, either untreated or treated according to the schedule depicted ($n$ = 16–27 for RIP1-Tag5 and $n$ = 5 for 4T1; **$P$ < 0.001 and ***$P$ < 0.0005 by log-rank Mantel–Cox test).

C       4T1 tumour-bearing mice were left untreated (UT) or treated with indicated compounds according to treatment schedule shown in panel (B). Top panel: Photographic images of excised 4T1 tumours from each treatment group. Bottom panel: Plots of individual tumour volumes (mm$^3$) and mean ± SEM on day 18 are shown ($n$ = 5–11; **$P$ < 0.01 by one-way ANOVA test with Tukey's correction).

D       Left: Representative flow cytometry plots of cytotoxic (Granzyme B$^+$) CD8$^+$ and CD4$^+$ T cells and regulatory (CD25$^+$, FoxP3$^+$) CD4$^+$ T cells isolated from 4T1 tumours of mice treated with 5 daily i.v. injections of CSG or 2 µg TNFα-CSG (gating strategies in Appendix Fig S3D). Graphs show mean ± SEM fractions of cytotoxic CD8$^+$ and CD4$^+$ T cells and regulatory CD4$^+$ T cells in each treated tumour ($n$ = 5; *$P$ < 0.05 and ***$P$ < 0.001 by Student's $t$-test).

Source data are available online for this figure.

for improved accumulation of contrast agents, enabling better MRI contrast.

The immune-enhancing effects of TNFα-CSG translated into therapeutic benefits as shown by reduced tumour growth and enhanced survival in tumour-bearing mice with an intact immune system. Importantly, this was not the case in mice with a compromised adaptive immune system. TNFα-CSG therapy was ineffective in

BALB/c nude mice, and transfer of normal CD8$^+$ and CD4$^+$ T cells into these mice restored the ECM degradation and anti-tumour effects. These findings show that the TNFα-CSG effects are primarily mediated by enhanced T-cell accumulation in solid tumours. These cells included effector T cells (granzyme B$^+$ CD8$^+$), which are indicative of anti-tumour immunity (Russell & Ley, 2002; Hoves *et al*, 2012).

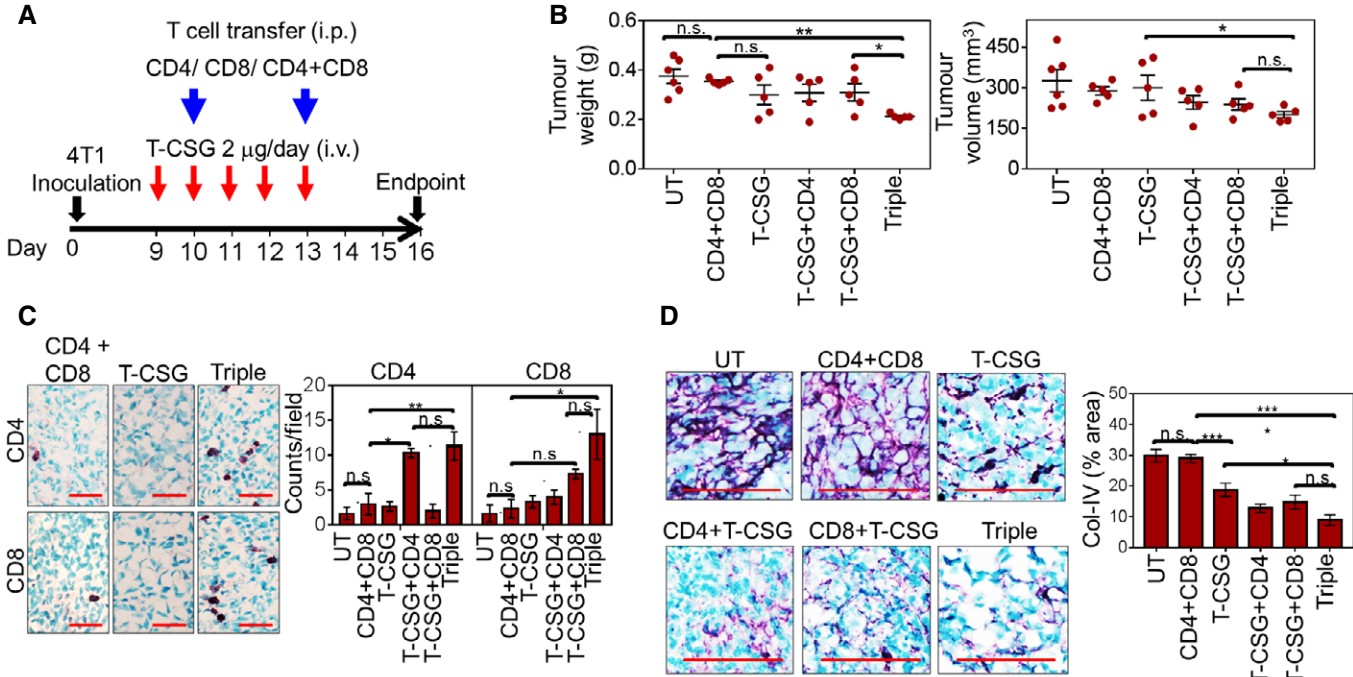

**Figure 8. CD8⁺ and CD4⁺ T cells mediate ECM degradation and anti-tumour effects of TNFα-CSG.**

A BALB/c nude mice bearing 4T1 tumours were treated with 5 daily i.v. injections of TNFα-CSG in the absence or presence of naïve splenic CD4⁺ and CD8⁺ T cells, or both (triple treatment). The cells were injected i.p. on days 10 and 13. On day 16 post-implantation, the tumours were analysed for weight and volume (B), immune infiltrates (C) and collagen IV content (D).

B Plots of individual tumour weights (g) and volumes (mm³) and mean ± SEM on day 16 are shown for untreated (UT) and treated groups ($n$ = 5; *$P$ < 0.05 and **$P$ < 0.005 by one-way ANOVA test).

C Left: Micrographs show CD4⁺ and CD8⁺ T cells (purple) counterstained with methyl green in tumours from the indicated treatment groups. Scale bars: 40 μm. Right: Bar charts show mean ± SEM of CD4⁺ and CD8⁺ T-cell counts/field ($n$ = 3; *$P$ < 0.05 and **$P$ < 0.005 by one-way ANOVA test with Tukey's correction).

D Left: Micrographs show representative collagen IV (Col-IV) expression (purple) counterstained with methyl green in untreated and treated tumours. Scale bars: 25 μm. Right: Bar charts show mean ± SEM of collagen IV staining/field ($n$ = 5; *$P$ < 0.05 and ***$P$ < 0.001 by one-way ANOVA test with Tukey's correction).

Source data are available online for this figure.

The ECM content and stiffness have been shown to influence the metastatic potential of tumours. For instance, stiff cancers can induce the expression of adhesion molecules which facilitate tumour cell adherence and transmigration through microvascular endothelia. Reducing these adhesive interactions by ECM degradation can thus be expected to prevent or reduce metastasis (Rath et al, 2017; Reid et al, 2017; Vennin et al, 2017). However, ECM degradation has also been shown to increase the release of tumour cells into the circulation, promoting metastasis (Sevenich & Joyce, 2014; Brown & Murray, 2015; Schmaus & Sleeman, 2015). Our data suggest that TNFα-CSG treatment, despite the resulting ECM loss, does not increase metastasis, potentially because it reduces tumour hypoxia. Metastasis often occurs under hypoxic conditions when tumour oxygen supply is compromised (Rahbari et al, 2016; Rankin & Giaccia, 2016).

TNFα is already a clinically approved agent (marketed by Boehringer Ingelheim as Beromun[R]). Because of the toxicity of TNFα, it is mainly applied locally, for instance in isolated limb perfusion to treat advanced melanoma and sarcoma (Lejeune et al, 2006). A few targeted TNFα compounds are in clinical trials; a TNFα fusion to an NGR-motif peptide, which targets tumour-associated aminopeptidase N (NGR-TNFα, Arenegyr;

MolMed), and an antibody-TNFα conjugate, which targets EDB domain-containing fibronectin in tumours (L19-TNF, Fibromun; Philogen), have entered multiple cancer trials (Gregorc et al, 2010; Spitaleri et al, 2013). However, these compounds primarily target tumour vessels (Pasqualini et al, 2000; Borsi et al, 2003), whereas our results show that CSG and its payload penetrate into ECM in extravascular tumour tissue, possibly because it engages the CendR transcytosis and transtissue transport pathway (Sugahara et al, 2009; Teesalu et al, 2009).

In summary, TNFα-CSG has the dual capacity to promote immune cell entry and ECM degradation within the tumour. Since the effects on immune cell infiltration and ECM depletion are intertwined, there are several potential clinical applications for TNFα-CSG in cancer treatment. For instance, pre-treatment with TNFα-CSG may sensitise inaccessible tumours for better detection as exemplified here by improved uptake of the nanoparticle contrast agent. Whilst we have not investigated the immunotherapeutic potential of TNFα-CSG, our findings suggest that TNFα-CSG may be used to inflame tumours which lack intrinsic effector T cells, and its full therapeutic potential could be achieved in combination with immune checkpoint blockade or other immunotherapeutic strategies.

# Materials and Methods

### Cell lines

Murine cell lines, including 6-thioguanine-resistant 4T1 breast cancer cells, colon carcinoma CT26 and MDA-MB-435 human carcinoma, were purchased from ATCC. Mouse brain endothelial cells (bEnd5) were kindly provided by B. Engelhardt (University of Bern, Bern, Switzerland). βTC-C3H tumour cells were derived from RIP-Tag mice bred on a C3H genetic background and were kindly provided by D. Hanahan (Institute Suisse de Recherches Experimentales sur le Cancer, Lausanne, Switzerland).

### Tumour-bearing mice

All mice were kept under specific pathogen-free conditions, and all experimental protocols were approved by the Animal Ethics Committees of the University of Western Australia (UWA), Sanford Burnham Prebys Medical Discovery Institute (SBP) and University of California, Davis (UCDavis). All mice were kept under specific pathogen-free facilities at UWA, SBP and UCDavis with food and water provided *ad libitum*. RIP1-Tag5 transgenic mice were established on a C3HeBFe background (kindly provided by D. Hanahan). Both male and female RIP1-Tag5 mice at 25–29 weeks of age were used in all studies. ALB-Tag transgenic mice were established on a C3HeBFe background and spontaneously develop hepatocellular carcinoma at 9 weeks of age. Both male and female ALB-Tag mice at 10–12 weeks of age were used in all studies. MTV-PyMT$^+$ mice were established on C57BL/6 background. Only 8-week-old female mice were used in the homing study. Unless otherwise stated, transplanted murine tumours were inoculated with $5 \times 10^5$ cells either orthotopically into mammary fat pad (4T1 and MDA-MB-435 cells) or subcutaneously (CT26 cells) in 9- to 12-week-old female mice, either syngeneic BALB/c (for 4T1 and CT26) or nude BALB/c (for MDA-MB-435 and 4T1). For adoptive transfer experiments, splenic CD4$^+$ and/or CD8$^+$ T cells isolated by MACS cell separation (Miltenyi Biotec) were injected into total of $2.5 \times 10^6$ cells per *i.v.* injection. All mice were randomly assigned to various experimental groups.

### Phage display

*In vitro* phage display was performed by incubating the phage library ($3.7 \times 10^{10}$ plaque-forming units (p.f.u.)) onto Matrigel™-coated tissue culture wells at 4°C overnight, and unbound phages were removed by serially washing with 1% BSA in DMEM. The bound phage was quantified by titration. The enriched phage pool from three rounds of *in vitro* biopanning on Matrigel™ was then subjected to four rounds of *in vivo* selection by injecting the Matrigel-selected library ($1.7 \times 10^9$ p.f.u.) into the tail vein of a BALB/c nude mouse bearing an MDA-MB-435 tumour followed by perfusion of the circulatory system and quantification of the bound phage in target organs by titration (Teesalu *et al*, 2012). Individual clones derived from this *in vivo*-selected tumour-homing pool were picked, and their peptide-encoding inserts were sequenced.

### Peptide synthesis

Peptides, amidated at the C-terminus, were synthesised on a microwave-assisted automated peptide synthesiser (Liberty; CEM,

Matthews, NC) following Fmoc/tertiary butyl strategy on rink amide MBHA resin with HBTU (N,N,N′,N′-tetramethyl-O-(1H-benzotriazol-1-yl)uronium hexafluorophosphate) activator (or alternatively, O-(benzotriazol-1-yl)-N,N,N′,N′-tetramethyluronium hexafluorophosphate), collidine activator base and 5% piperazine for deprotection. When included, fluorescein was incorporated during the synthesis at the N-terminus of the sequence as 5(6)-carboxyfluorescein with a spacer, 6-aminohexanoic acid, separating the fluorophore and the sequence. All the peptides were purified to purities > 90% by HPLC with acetonitrile and water gradient mixtures with 0.1% TFA. Peptides were analysed by MALDI and found to have [M + H] 1022.9 for Ac-CSG, 1451.853 for FAM-CSG, 1612.17 for FAM-ARALPSQRSR (FAM-ARA) and 1076.244 for FAM-CREKA.

### Production of recombinant proteins

Recombinant TNFα was produced as described previously (Johansson *et al*, 2012). Briefly, mature murine TNFα with or without a C-terminal-modified CSG peptide (CSGRRSSKC, connected via a GGG linker) was cloned into Xho/BamH1 sides of the vector pET-44a (Novagen) to express soluble fusion proteins with N-terminal Nus•Tag/His•Tag. Briefly, after isopropyl-β-D-glactopyranoside (IPTG) induction overnight at 25°C, cultures were centrifuged, resuspended in lysis buffer (50 mM NaH$_2$PO$_4$, 300 mM NaCl, 10 mM imidazole, 1 mM DTT, 1 mM PMSF, 1 mM EDTA and 1% Triton X-100 (vol/vol) at pH 8.0), incubated with 1 mg/ml lysozyme on ice for 30 min, sonicated and purified using Ni-NTA beads (Qiagen) according to the manufacturer's instructions. Nus•Tag/His•Tag was cleaved with tobacco etch virus (TEV) protease at 30°C for 6 h. Recombinant proteins from cleavage reactions were dialysed overnight in PBS and purified by using Ni-NTA beads. Purity was assessed on Coomassie Brilliant Blue-stained protein gels. The recombinant TNFα-CSG and unconjugated TNFα are approximately 18.9 and 17.7 kDa.

### *In vitro* and *in vivo* peptide binding

Peptide binding *in vitro* was assessed by incubating 5 μM FAM-CSG, FAM-ARA or FAM-CREKA on 8-μm tissue cross-sections for 30 min, and tissues were analysed by microscopy. For *in vivo* biodistribution analysis, tumour-bearing mice were injected intravenously (tail vein) with 100 μl of 1 mM FAM-CSG or FAM-ARA. After 1 h, animals were euthanised and tissues, including tumour, heart, liver, spleen, kidney, skin, muscle, brain, lung, small intestine, vertebrae and pancreas, were excised and imaged under a UV illuminator (Illumatool, Lightools Research, CA, USA) for green fluorescence intensity. Tissues were embedded in O.C.T. (Tissue-Tek®) as fresh frozen and stored at −80°C for further histology analysis.

### Peptide binding on human tissues

All experimental protocols were approved by Human Ethics Committees of the University of Western Australia and Sir Charles Gairdner Hospital. Informed consent was obtained from all subjects and that the experiments conformed to the principles set out in the WMA Declaration of Helsinki and the Department of Health and Human Services Belmont Report. Breast tumour and adjacent samples > 1.5 cm were obtained as fresh and unfixed specimens

within 1 h post-mastectomy from consenting $n = 7$ breast cancer patients. The breast cancer biopsies were classified as stage 2–3 (grade 3) carcinoma ($n = 2$) and invasive ductal carcinoma ($n = 2$), all within 35–50 mm in diameter. The patients did not receive any treatment prior to the mastectomy. Pancreatic ductal carcinoma (PDAC) samples were obtained from $n = 3$ untreated patients following pancreaticoduodenectomy. Hepatocellular carcinoma (HCC) samples were obtained from $n = 3$ HCC patients undergoing hepatectomy. In binding studies, acetone-fixed serially cut 8-µm tissue sections of untreated specimens were first blocked with 5% BSA solution and incubated with 1 µM FAM-CSG or control FAM-ARA for 30 min, and washed three times with PBS for 5 min. In CSG blocking studies, the tissue sections were pre-incubated with 1 mM unlabelled CSG peptide for 20 min prior to incubation with FAM-CSG. Peptide binding was determined based on the tissue reactivity to anti-FITC-HRP antibody.

## Affinity chromatography for receptor isolation

Matrigel extract at 0.5 mg/ml in buffer containing 50 mM Tris, 5 mM EDTA-Na and protease cocktail inhibitor (Thermo Scientific) was incubated with CSG-coated Sulfolink beads (Pierce Biotechnology) at 4°C. After washing, bound proteins were eluted with the same buffer containing 2 mmol/l free CSG or CREKA peptide and separated by sodium dodecyl sulphate–polyacrylamide gel electrophoresis. Gel bands excised from silver-stained gels were digested, and peptides were extracted as described previously (Bringans *et al*, 2008). Samples were analysed an Agilent 6540 Q-TOF mass spectrometer (Agilent Technologies) with an HPLC Chip Cube source. The analyses were performed in the WA Proteomics Facility, supported by Lotterywest and Bioplatforms Australia, at the Harry Perkins Institute for Medical Research.

## TNFα-CSG homing and treatment studies

To investigate intratumoral accumulation of TNFα-CSG, 100 µg of FITC-labelled TNFα-CSG or TNFα (via amine-reactive NHS–fluorescein conjugation, Thermo Scientific) was i.v. injected into tumour-bearing mice. After 1 h, animals were euthanised and tissues, including tumours, heart, liver, kidney and pancreas, were excised, embedded in O.C.T. and stored at −80°C for further histological analysis. In treatment studies, RIP1-Tag5 and 4T1 tumour-bearing mice were i.v. injected with PBS, CSG, TNFα, TNFα-CSG or TNFα-RGR at required doses. Analyses of lectin and IONP uptake, FACS, micro-elastography and DCE-MRI were done within 24 h after final treatment was administered to the tumour-bearing mice, unless otherwise stated.

## Lectin perfusion and hypoxia assessment

For lectin perfusion, the mice were i.v. injected with 100 µg of FITC-labelled tomato lectin (Lycopersicon esculentum; Vector). After 10 min of circulation, mice were heart-perfused with 2% neutral-buffered formalin (w/v) and tumours were frozen in O.C.T. compound. To evaluate tumour hypoxia, mice were i.v. injected with 1.5 mg pimonidazole (Hydroxyprobe™-1). Tissues were collected after 1-h circulation, fixed with 2% neutral-buffered formalin and frozen in O.C.T. after 2 h in 10 and 30% sucrose gradients.

## Histology analysis

Tissue distribution of fluorescein (FAM/FITC)-labelled peptides and recombinant proteins was detected on 8-µm tissue cross-sections (prepared from O.C.T. embedded tissues) based on their fluorescence intensity and reactivity to anti-FITC-HRP antibody (polyclonal, Gentex, 1:300) and counterstained with haematoxylin or methyl green (Vector Laboratories). For immunohistochemistry, the following antibodies were used: anti-CD8a (53-6.7; eBioscience, 1:100), anti-CD4 (GK1.5; eBioscience, 1:50), anti-CD45 (30-F11, eBioscience, 1:100), anti-CD31 (390; eBioscience, 1:100), anti-CD106 (VCAM-1, 429; eBioscience, 1:100), anti-collagen IV (polyclonal; Abcam, 1:200), anti-CD11b (M1/70; BD Pharmingen, 1:100), anti-nidogen-1 (ELM1; Millipore or polyclonal; R&D Systems, 1:50), anti-laminin (polyclonal; Millipore, 1:300), anti-fibronectin (polyclonal; Abcam, 1:100), anti-heparan sulphate proteoglycan 2 or perlecan (A7L6; Abcam, 1:100), anti-collagen I (polyclonal; Abcam, 1:500), ER-TR7 (murine thymic reticulum; Abcam, 1:100) and anti-Ki67 (SolA15; eBioscience, 1:100). For secondary detection, Vector VIP Peroxidase Substrate Kit (Vector Laboratories) as well as fluorescence-labelled, Alexa Fluor 488- or 594-conjugated anti-rat, and anti-rabbit or anti-goat (Life Technologies) were used. Tissues from pimonidazole-injected mice were stained with anti-pimonidazole (Hydroxyprobe™-1) based on the product standard protocol. Trichome connective tissue staining, picrosirius red staining (Abcam) and TUNEL assay (Sigma) were performed as per product protocol. Images were captured on a Nikon Ti-E microscope (Nikon Instrument Inc, N.Y., USA) or ScanScope XT (Aperio Technology, Inc, CA, USA). Image analysis and quantification were performed using either NIS software modules (version 4.0) or ImageScope version 12.1.0.5029 (Aperio Technology, Inc, CA, USA).

## Isolation of immune cells and flow cytometry analysis

4T1 tumours were excised from CSG and TNFα-CSG-treated mice, minced and incubated in DMEM high-glucose medium containing 0.1 mg/ml DNase I (Sigma) and 0.5 mg/ml collagenase IV (Sigma) for 1 h at 37°C under gentle rotation. The cell suspension was passed through a 70-µm membrane and subsequently washed with FACS buffer [1% BSA (w/v, Sigma) in PBS]. For analysis of immune cells, viable cells (Zombie Aqua™ selection; BioLegend) were stained with the following antibodies: CD45-APC-efluor 780 (30-F11; eBioscience, 1:200), CD3-BUV395 (145-2C11; BD, 1:200), CD4-BV510 (Gk1.5; BD, 1:100), CD8-PE-CF594 (53-6.7; BD, 1:300), CD11b-PE (M1/70; BD, 1:300), F4/80-AF647 (BM8; BioLegend, 1:300) and Ly6G-FITC (1A8; BD, 1:100). For analysis of cytotoxic and regulatory T cells, viable cells were stained with the following antibodies: CD3-APC-eFluor 780 (145-2C11; eBioscience, 1:200), CD4-FITC (Gk1.5; eBioscience, 1:200), CD8-PerCp-eFluor 710 (53-6.7; eBioscience, 1:200), CD25PE-efluor 610 (PC61.5; eBioscience, 1:100), FoxP3-AF647 (150; BioLegend, 1:100) and granzyme B (GB11; BD, 1:100). For intracellular staining (granzyme B and FoxP3), cells were permeabilised with True-Nuclear™ Transcription Factor Buffer (Biolegend). Cells were sorted on a BD SORP Fortessa and analysed on BD FACSDiva software version 8.0.1 (BD Biosciences, USA).

## Protease gene expression analysis of tumour-infiltrating immune cells

4T1 tumours were excised from CSG and TNFα-CSG-treated mice, minced and incubated in DMEM high-glucose medium containing 0.1 mg/ml DNase I (Sigma) for 30 min at 37°C under gentle rotation and passed through a 70-μm membrane. CD45$^+$/CD11b$^+$ and CD45$^+$/CD11b$^-$ cells were sorted on a FACS Aria II (BD Biosciences), and RNA from each sample was prepared using the RNA Easy Mini Kit (Qiagen). For cDNA synthesis and qPCR, the RT$^2$ system (Qiagen) was used according to the manufacturer's protocol. The gene expression profiles for urokinase plasminogen activator (uPA), matrix metalloproteinases (MMPs, including MMP2, MMP3, MMP9, MMP12 and MMP14), cathepsins (B, L, S and C) and ADAMs (8, 9, 10 and 17) were assessed using a Q-PCR Kit (Qiagen) on the Rotor-Gene Real-Time PCR Detection System. All reactions were normalised to hypoxanthine-guanine phosphoribosyltransferase (HPRT). Primer sequences include uPA F (CGATTCTGGAG GACCGCTTA), uPA R (GATTGAATCCAGTCCAGGAAGT), MMP2 F (CCTGGACCCTGAAACCGTG), MMP2 R (TCCCCATCATGGATTC GAGAA), MMP3 F (CCCCTGATGTCCTCGTGGTA), MMP3 R (GCA CATTGGTGATGTCTCAGGTT), MMP9 F (GGACCCGAAGCGGACAT TG), MMP9 R (GAAGGGATACCCGTCTCCGT), MMP12 F (ATGTG GAGTGCCCGATGTACA), MMP12 R (AAGTGAGGTACCGCTTCA TCCAT), MMP14 F (CATCATGGCCCCCTTTTACC), MMP14 R (CGATCGTCATCAGGCAACAC), cathepsin B F (TTTGATGCACGGG AACAATG), cathepsin B R (TTGGTGTGAATGCAGGTTCG), cathepsin C F (AAATGAGCCTGCGAGATCTGA) cathepsin C R (ATTGA CGCCTTGGACGTTTC), cathepsin L F (AATACAGCAACGGGCA GCA), cathepsin L R (AGCGGTTCCTGAAAAAGCCT), cathepsin S F (CGCCAGCCATTCCTCCTT), cathepsin S R (ATGATTCACATTG CCCGTACA), ADAM 8 F (CACCACTCCCAGTTCCTGTT), ADAM 8 R (AAGGTTGGCTTGACCTGCT), ADAM 9 F (CCTTGCCTCTCTGCG ACTAA), ADAM 9 R (AATTTCATAAGAAGAAAGATGGACAG), ADAM 10 F (TTATGCCATGTTTGCTGCATGA), ADAM 10 R (ACCACTGAACTGCTTGCTCCAC), ADAM 17 F (CAGCAGCACTCCA TAAGGAA) and ADAM 17 R (TTGTGAAAAGCGTTCGGTA).

## Measurement of protease levels

Spleens were excised from RIP1-Tag5 mice, minced and incubated in RPMI for 30 min at 37°C under gentle rotation and passed through a 70-μm membrane. CD4$^+$ and CD8$^+$ T cells and CD11b$^+$ monocytes/macrophages were isolated by MACS cell separation (Miltenyi Biotec). Cells (300,000 cells/well) were seeded onto Matrigel embedded with 0.4 μg CSG or 1 μg TNFα-CSG, and were incubated with TexMACS™ (Miltenyi Biotec, T cells only) or RPMI (monocytes). The levels of proteases secreted in the supernatant at 4 and 20 h after seeding were measured by ELISA following the manufacturer's instructions. ELISA kits used include mouse MMP9 (BosterBio), MMP12 (Abcam) and cathepsin L (Cloud-Clone Corp).

## Measurement of plasma CRP

Healthy C3H mice were treated with either (i) a single injection of PBS, TNFα-CSG or TNFα or (ii) 5 daily injections of TNFα-CSG or CSG at required doses. Plasma samples were collected retro-orbitally, 5 h after a single-dose treatment and 48 h after the mice were treated with 5 daily doses of TNFα-CSG and CSG. Measurement of plasma CRP was performed by the PathWest Clinical Biochemistry Laboratory, Queen Elizabeth II Medical Centre. Plasma CRP was measured on a routine clinical chemistry analyser, Abbott Architect c16000, by immunoturbidimetry, with a detection limit of 0.2 mg/l and a measuring range of 0.2–480 mg/l (Abbott Laboratories, Abbott Park, Illinois).

## 4T1 lung metastasis assay

Spontaneous 4T1 lung metastases in response to TNFα-CSG and CSG treatment were measured following a published procedure (Pulaski & Ostrand-Rosenberg, 2001). Briefly, on day 19 post-inoculation (in which most tumours in CSG-control group reached the ethical size endpoint of 1,500 mm$^3$), 0.7 ml blood samples, lungs, liver and sentinel lymph nodes were removed from TNFα-CSG and CSG-treated tumour-bearing mice. Tissues were finely minced and digested in 1× HBSS containing 1 mg/ml collagenase IV and 6 U/ml elastase for 75 min at 4°C under gentle agitation. Digested cells were washed, filtered and plated in serial dilutions on 10-cm culture dishes containing growth medium and 60 μM 6-thioguanine. After 10 days, 6-thioguanine-resistant colonies were fixed in methanol and stained with 0.03% (w/v) methylene blue. Methylene blue$^+$ cells were imaged on a Nikon Ti-E microscope. For each sample, the average fraction of 4T1 cells$^+$/field of view from a minimum of 15 fields of view was quantified using NIS software modules (version 4.0).

## Quantitative micro-elastography

Freshly isolated 4T1 tumours from mice treated with TNFα-CSG were subjected to quantitative micro-elastography, an optical coherence tomography (OCT)-based elastography method described previously (Kennedy et al, 2014, 2015a,b) to produce maps of the stiffness of a tissue with microscale resolution. The imaging system comprises a custom-built OCT system and a piezoelectric actuator-based mechanical loading apparatus. The spectral-domain OCT system operates at a central wavelength of 840 nm, with a measured axial resolution of 8 μm and a measured lateral resolution of 11 μm. Each tumour was covered with a transparent compliant silicone layer with known mechanical properties (P7676, Wacker, Germany) and compressed between a glass window and a compression plate. A piezoelectric ring actuator (Piezomechanik, Germany), affixed to the imaging window, was used to apply a force (1–10 newtons) to the tissue. The force induced a micrometre-scale deformation within the sample, and the axial component of this deformation was measured using phase-sensitive OCT. Local axial strain at each location with the OCT field of view was calculated as the change in axial displacement with respect to depth over 100 μm within the tissue. The stress imparted at the tissue surface by the actuator was estimated by measuring the strain in the compliant layer and converting it to stress via a stress/strain look-up table for the silicone material. Maps of tumour stiffness, $E$, were generated by dividing the local stress, $\sigma$, by the local strain, $\varepsilon$, i.e., $E = \sigma/\varepsilon$. Stiffness variance, a measurement of the spread of tumour stiffness (range from 1–1,000 kPa), was calculated as the average squared deviation of each number

from its mean $\left(\frac{\sum (x-\mu)^2}{N}\right)$ where $x$ = stiffness of individual pixel, $\mu$ = mean of stiffness, and $N$ = total number of pixels).

### In vivo DCE-MRI

4T1 tumour-bearing mice, treated with TNFα-CSG or CSG, were scanned on a Bruker BioSpec 70/30 (7T) small animal scanner (Bruker BioSpin MRI, Ettlingen, Germany). Gadoteridol (Bracco, Singen, Germany) was administered approximately 90 s after the start of the scan (0.3 μmol/g, i.v. bolus), and data were collected for ~10–15 min. Data were acquired, and images were reconstructed using ParaVision 5.1 (Bruker BioSpin MRI) as previously described (Thorsen et al, 2013). Parametric images were generated with either ParaVision 5.1 or MATLAB (MathWorks). A T1 map (RAREvtr: echo time (TE) = 22 ms; repetition time (TR) = 5,000, 3,000, 1,500, 1,000, 800, 500, 365 ms; field of view (FOV) = $1.8 \times 3.6$ cm$^2$; matrix (MTX) = 64 × 128; slice thickness (ST)/slice interval (SI) = 1 mm/1 mm; and 9 slices) was performed prior to the administration of gadolinium contrast for later use in deriving the gadolinium contrast agent concentration from the MR signal intensity. A series of T1-weighted gradient-echo images was acquired (FLASH; TE/TR/flip angle (FA): 2.7 ms/100 ms/30°; FOV = 4 × 2 cm; matrix = 160 × 80; 11 slices; and 100 repetitions) with a temporal resolution of 8 s. A ROI was drawn around the tumour, and the mean signal intensity versus time was converted to gadolinium concentration versus time using: (i) a T1 map acquired prior to the DCE image series and (ii) the previously measured relativity of gadoteridol in whole mouse blood. The contrast agent versus time curve was used as an input in a two-compartment pharmacokinetic model, and a fit was obtained using pmod (PMOD Technologies Ltd, Zurich, Switzerland) software.

### Preparation of iron oxide nanoparticles (IONP)

Iron oxide (Fe$_3$O$_4$) was prepared by a precipitation method, as described previously (Kang et al, 1996). The bare iron oxide was then coated with dextran (MW 5,220, Sigma, St. Louis, MO, USA) as described (Yu et al, 2012), generating particles of ≈13–18 nm in diameter. Fluorescein lipid labelling was prepared by coupling 1,2-distearoyl-sn-glycero-3-phosphoethanolamine-N-maleimide (DSPE-PEG$_{2000}$-maleimide, Avanti Polar Lipids) with carboxyfluorescein (FAM) bearing a cysteine on its N-terminus in 1:1 molar ratio. The dextran-coated iron oxides were encapsulated with PEGylated lipids by combining dextran-coated iron oxides (in hexane) with DSPE-PEG$_{2000}$ and DSPE-PEG$_{2000}$-FAM (in chloroform) in 1:1:0.37 molar ratios (Sugahara et al, 2009; Starmans et al, 2013). The final coated IONP was 33 ± 3 nm in diameter (zeta potentials = 32.20 ± 2.26 mV), as measured by dynamic laser light scattering on a Malvern Zetasizer Nano ZSP (Malvern, U.K.).

### Ex vivo MRI and histological detection of intratumoral IONP uptake

FAM-labelled IONP (5 mg/kg) were i.v. injected into 4T1 and RIP1-Tag5 tumour-bearing mice treated with TNFα-CSG or CSG. After 4-h in vivo circulation and heart perfusion with PBS to

### The paper explained

#### Problem
Most cancers are stiffer than normal tissue because they produce an abundance of extracellular matrix (ECM), which forms a "barrier" for drugs or immune cells. ECM depletion is a compelling strategy to reduce tumour stiffness, decompress tumour blood vessels and thus improve access of diagnostic and therapeutic agents, and anti-tumour immune cells to the cancer core. However, it remains challenging to ensure tumour ECM-targeting specificity with high local activity and minimal toxicity to normal organs.

#### Results
In this study, we use a new biological agent, so-called TNFα-CSG, an immune-modulating cytokine that is specifically targeted to tumour ECM. The peptide-targeting moiety in this agent is highly specific for mouse and human cancer ECM and does not bind to normal tissue. TNFα-CSG treatment in pre-clinical cancer models attracts immune cells which in turn secrete a cocktail of proteases and degrade ECM locally. Reduced tumour stiffness and improved tumour perfusion enhance tumour uptake of nanoparticles/imaging agents. Therapeutically, TNFα-CSG is highly effective by activating anti-cancer T cells and suppresses tumour growth without systemic toxicity or increase in metastatic activity.

#### Impact
This study demonstrates the local and selective activity of a new ECM-targeting biological agent which reduces cancer stiffness and increases perfusion. The novel combination of matrix depletion with immune modulation facilitates drug and immune cell access into fibrotic tumours and warrants clinical investigation.

remove circulating particles, tissues were collected and incubated in 2% formalin for 2 h and embedded in 2% agarose. MRI scans were performed at 9.4 T with a Bruker BioSpec 94/30 magnet, Avance III HD console (Bruker BioSpin GmbH, Ettlingen, Germany). Images were visualised by ParaVision 6.0.1 acquisition software. Following a localiser, a series of three scans was completed with 23 interleaved coronal slices: (i) a T2*-weighted image [FLASH: echo time (TE) = 10 ms; repetition time (TR) = 600 ms; flip angle (FA) = 30°; averages (NA) = 2; and acquisition time (TA) = 3 min]; (ii) T2* map [MGE: TR = 1250 ms; TE = 2.6 ms; FA = 90°; number of echoes (NE) = 10; echo spacing (ESP) = 5 ms; TE (effective) = 2.6–47.6 ms; and TA = 5 min]; and (iii) a T2 map [MSME: TR = 3150 ms; TE = 8 ms; FA = 90°; refocussing pulse = 180°; NE = 16; ESP = 8 ms; and TA = 7 m 53 s]. For above scans, field of view (FOV) = $3.6 \times 3.6$ cm$^2$, matrix (MTX) = 240 × 240, slice thickness (ST) = 0.5 mm, and slice gap = 25 μm. The T2* and T2 parameter maps for each tumour were calculated from the MGE and MSME datasets, respectively, using the ParaVision 6.0.1 macro fitinlsa that fits the signal for each pixel according to a mono-exponential decay. Using the Image Display and Processing Tool, tumour borders were segmented manually using the track tool, based on the MSME image with TE (effective) = 8 ms. Subsequently, the regions of interest (ROI) were used in combination with the parameter maps to obtain statistics on the T2* and T2 values for each ROI. Statistics for tumours were obtained by combining the ROI statistics on an image slice by slice basis. Tissue distribution of fluorescein-labelled IONPs was detected on 8-μm tissue cross-sections based on their reactivity to

anti-FITC-HRP antibody (polyclonal, Gentex) and counterstained with haematoxylin.

### Statistical analysis

No statistical methods were used to predetermine sample size. A minimum of 3 mice per group was used in all studies. However, since each RIP1-Tag5 mouse usually carries multiple tumour nodules (2–6 tumours/mouse), all graphs for RIP1-Tag5 mice show data points from 2 to 6 tumours per mouse. For histological quantification, raw data were obtained from each microscopically identified tumour on tissue section (2–4 identified tumours/tissue sections). Raw data were obtained from at least three fields of view for tumours > 3 mm or from the entire tumour section for tumours < 3 mm. All studies were not blinded. For survival study of 4T1 tumour-bearing mice, age-matched animals with similar tumour burden were randomised and treated in groups of 4–5 mice. Survival of RIP1-Tag5 tumour-bearing mice was based on welfare impact scores and independently evaluated as part of technical service at the animal facility. Statistical analyses were performed using GraphPad Prism 7 (GraphPad Prism Software, Inc.). Data were analysed by Student's *t*-test (two-tailed) or one-way analysis of variance (ANOVA) unless indicated otherwise. Cumulative survival time was calculated by the Kaplan–Meier method and analysed by the log-rank test. A *P* value < 0.05 was considered statistically significant. Error bars indicate SEM. Experiments were carried out in an unblinded fashion.

Expanded View for this article is available online.

### Acknowledgements

We thank Prof. Evan Ingley and Dr. Anna Johansson-Percival (Harry Perkins Institute) and Prof. Greg Sterrett (PathWest, Sir Charles Gairdner Hospital) for providing reagents or help with tissue analysis. This work was supported by Cancer Council WA grants to J.H.; NHMRC Career Development Fellowship to J.H. (APP1046507); NHMRC project grants to J.H., R.G. and E.R. (APP1058073, APP1121131); CA188883 from the US National Cancer Institute to E.R.; and NIHCA210553 to K.W.F. The authors acknowledge the facilities and the scientific and technical assistance of the National Imaging Facility at the Centre for Microscopy, Characterisation & Analysis, UWA, a facility funded by the University, State and Commonwealth Governments.

### Author contributions

JH, RG and ER designed and planned the experiments, interpreted findings and wrote the manuscript. JH, YLY, VKR, TDS, XW, NA, JW, DSD, HB, IL, TF, APM and JL planned and carried out synthesis, purification and biological characterisation of reagents. JH, MC, KF, LMM, BZF, HZ, FAA, MPK, MJ, HCE, GY, LM, KWF, KMK, WMA, BFK and DDS planned, carried out and interpreted imaging studies.

### Conflict of interest

OncoRes Medical Pty Ltd is developing optical elastography for applications in breast-conserving surgery. B.F.K., K.M.K and D.D.S. have shares in OncoRes Medical, and B.F.K. is undertaking funded research for this company. J.H., R.G. and E.R. filed a patent on the biomedical applications of CSG and TNFα-CSG. (patent title: "Novel biomolecule conjugate and uses therefor", international application number: PCT/AU2017/050037; applicants: University of Western Australia and Sanford Burnham Prebys Medical Discovery Institute.)

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
