## [Review Process File · EMBO Molecular Medicine]

Immune-mediated ECM depletion improves tumour perfusion and payload delivery

Yen Ling Yeow, Venkata Ramana Kotamraju, Xiao Wang, Meenu Chopra, Nasibah Azme, Jiansha Wu, Tobias D. Schoep, Derek S. Delaney, Kirk Feindel, Ji Li, Kelsey M. Kennedy, Wes M. Allen, Brendan F. Kennedy, Irma Larma, David D. Sampson, Lisa M. Mahakian, Brett Z. Fite, Hua Zhang, Tomas Friman, Aman P. Mann, Farah A. Aziz, M Priyanthi Kumarasinghe, Mikael Johansson, Hooi C Ee, George Yeoh, Lingjun Mou, Katherine W. Ferrara, Hector Billiran, Ruth Ganss, Erkki Ruoslahti, Juliana Hamzah

Review timeline:

Submission date:	27 May 2019
Editorial Decision:	20 June 2019
Revision received:	15 August 2019
Editorial Decision:	4 September 2019
Revision received:	26 September 2019
Accepted:	9 October 2019

Editor: Céline Carret

Transaction Report:

1st Editorial Decision

20 June 2019

Thank you for the submission of your manuscript to EMBO Molecular Medicine. We have now heard back from the three referees whom we asked to evaluate your manuscript.

You will find below the referees' comments. You will see that overall, the reports are encouraging. All referees request additional data to increase the conclusiveness of the findings, *in vitro* staining, and different analyses are suggested but also *in vivo* experiments (maybe using different models would strengthen the conclusions), and add more patients' samples (ref.1). While mechanism and clinical insights are requested, at this stage, we would like you to focus on making the paper stronger by addressing all issues pertaining to the clinical and translational effects. We would be happy to extend our revision time to 4 months.

We would therefore welcome the submission of a revised version within three months for further consideration and would like to encourage you to address all the criticisms raised as suggested to improve conclusiveness and clarity. Please note that EMBO Molecular Medicine strongly supports a single round of revision and that, as acceptance or rejection of the manuscript will depend on another round of review, your responses should be as complete as possible.

I look forward to receiving your revised manuscript.

***** Reviewer's comments *****

Referee #1 (Comments on Novelty/Model System for Author):

I suggest to exploit preclinical models nearer to human cancers where fibrosis represents a real obstacle for the drug delivery

Referee #1 (Remarks for Author):

In this paper the authors describe a new biotechnological tool designed to precisely deliver TNF to the tumor ECM, trigger an inflammatory reaction and dampen the tumor growth. To reach this aim TNF is linked to the CSG peptide identified by screening a phage library on ECM (Matrigel). The experiments were mainly performed in an insulinoma transgenic model (RIP-Tag) and in a triple negative breast cancer model (4T1). In my opinion the authors' hypothesis should be validated in models, which better recapitulate the clinical need to overcome the effect of fibrosis on drug delivery, such pancreatic ductal adenocarcinoma.

The data shown in Fig 1 seem to support the specificity of CSG peptide in detecting tumor ECM. The authors also demonstrate that it binds molecular determinants of basal membrane including laminin, nidogen-1 and collagen IV. The use of Matrigel to identify tumor ECM is fine because it is produced by a murine sarcoma. However this section has to be improved by a more precise identification of the molecule(s) identified by CSG. Furthermore the ECM of other pathological tissues besides cancer (eg lung, liver, wounded tissues) has to be tested. The demonstration of the capability of CSG to bind human cancers is just based on 4 breast cancers and therefore is weak. I suggest to extend the analysis to more patients with breast cancer, prostate cancer, pancreatic ductal adenocarcinoma and HHC.

Finally, the authors have to provide a first analysis of CSG binding to normal and cancer cells.

Fig 2. Besides the evaluation of the immune-infiltrate by immunofluorescence, a more precise evaluation of immune-profile by FACS is required. There are 3 other points: i) the type of polarization of the recruited macrophages; ii) the systemic effect of CSG-TNF: does it affect the feature of circulating or splenic lymphomononuclear cells? iii) the presence of polymorphonuclear cells in the treated tumors

Fig 3. Besides specific staining with Abs, a general view of ECM by Masson's trichrome and Sirius red stains should be useful. Furthermore, besides the two exploited models, I think that the best one to support the working hypothesis of the authors is to test CSG-TNF on a PDAC transgenic or orthotopic model, which are used to be characterized by a frank desmoplastic reaction.

In the same figure the authors show an inverse correlation between the expression of collagen and the CD45 recruitment. Is it just determined by a decompressing effect and an increased diapedesis of circulating CD45 cells? Or by the induction of a new and precise genetic program characterized by the appearance of chemotactic molecules?

Which is the effect of CSG-TNF on the presence of myofibroblasts/CAFs?

Fig 2 -4. Are the phenotypes here described reverted along the time and after the treatment interruption?

Fig 4 clearly shows that CSG-TNF modify the tumor stiffness. Is there any correlation between this effect and the proliferative and apoptotic index of cancer cells? Can the authors mimic the in vivo results by analyzing the in vitro behavior of cancer cells maintained on matrixes with different stiffness?

Fig 7. Do CD8 cells isolated from treated tumors exert an in vitro cytotoxic effect on cancer cells?

Referee #2 (Remarks for Author):

The manuscript by Yeow et al. describes a potentially highly interesting approach, based on a tumour ECM-homing CSG peptide coupled to the cytokine, tumour necrosis factor alpha, to induce immune cell infiltration and ECM degradation. The consequences of desmoplastic tumor microenvironment for tumor aggressive properties and therapy resistance are being increasingly appreciated. Therefore, by providing means to dissolve the desmoplastic ECM, the reported approach could have valuable clinical implications for the treatment of different cancers. However, there are some concerns and questions which should be addressed in order to strengthen the manuscript.

Specific comments:

1. A major structural ECM component in the desmoplastic tumour ECM is collagen I, in addition to other fibrillar collagens, fibronectin etc, which co-localization with FAM-CSG, and degradation/loss after the treatments should be carefully addressed. This is a critical point to understand the biological and possible clinical relevance of the developed tumour targeting approach: Therefore, in human and mouse tumors, FAM-CSG binding/homing should be compared to interstitial collagen and fibrillar ECM structures of the desmoplastic tumor microenvironment rather than the basement membrane components only (in addition to immunohistochemistry e.g. Trichrome stain could be used).
2. Figure 1B: What are the FAM-CSG+ round structures in 4T1 T; do they represent basement membranes, fibrillar ECM or something else? Are the FAM-CSG+ areas in RT5 T acellular? More detailed histological images and result description will allow better consideration of the specificity of CSG homing. Co-localization should be compared between different types of ECM components such as the BM proteins, fibrillar collagens and fibronectin. Double staining with markers for cancer-associated fibroblasts and/or tumor cells would also help in defining more specifically the targeted structures in tumors/tumor-associated stroma.
3. The authors state that CSG homing is specific for ECM (e.g. in the following citation from Abstract): "This peptide specifically targets tumour ECM of mouse and human carcinomas and selectively delivers a recombinant TNF α -CSG fusion protein to tumour ECM in tumour-bearing mice." However, cellular localization patterns are also shown in e.g. Fig S1G. How was cellular homing and delivery addressed/ruled out? Could cell dissociation followed by FACS be used to show the binding or lack of binding/delivery to cells?
4. Figure 1E shows a pattern of specific CSG binding in one human breast tumor section. Will similar ECM-patterns be seen in majority of human breast cancers and other types of human cancer?
5. Figure 3: The loss/degradation of proteins like collagen I/III/V and fibronectin which are major components of the desmoplastic tumour ECM should be compared and quantified in addition to the basement membrane proteins analyzed. Collagen IV is a network-forming collagen and a component of basement membranes; therefore it is unclear what is the relevance of reduced Col-IV (or the other BM proteins), used as the main read-out throughout the manuscript, for tumour stiffness and physical properties.
6. Does the degradation of (vascular) basement membranes affect vascular leakage or the amount (area) or proliferation of the CD31+ vascular cells?
7. The authors mention in discussion that CSG contains a potential CendR motif for transcellular and trans-tissue transport pathway. It would be important to address, if the presented peptide targeting/homing pattern is due to the specific ECM binding capability of CSG or if it involves the transcellular and trans-tissue transport pathway.
8. All the presented results and discussion considered, it is difficult to draw concise conclusions about the various aspects of the reported phenotype with regards to the underlying molecular mechanisms or the most relevant clinical implications. For example, is the "therapeutic benefit" of TNF α -CSG treatment dependent on the altered ECM (or vascular) phenotype, or on the immune cell influx only? Were the immune cell influx, decreased tumour growth and proliferation associated with increased tumor cell killing/death?

9. The importance of the ECM degradation for the described TNF α -CSG treatment outcome could be investigated by testing the effects of relevant protease inhibitors on tumor growth, immune cell influx and vascular perfusion. Or if the vascular phenotype including increased perfusion suggestive of conditions for improved drug delivery was a key aspect of the TNF α -CSG treatment-induced phenotype, a proof-of-principle combination treatment with relevant anti-cancer drugs would strengthen the study.

Minor points:

Which of the several laminin(s) were analyzed?

Figure 1E: Imaging/image processing; the background in Hu N FAM-CSG and FAM-ARA appear darker than in the other samples.

Referee #3 (Remarks for Author):

In this manuscript, the authors describe how a targeted form of TNF α can increase the infiltration of tumor-associated extracellular matrix (ECM) by immune cells. The targeting sequence was identified by screening phage-displayed peptide libraries with an *in vivo* approach that was based on selective tumor recognition. As a consequence, the ECM becomes degraded and less stiff, allowing for greater tumor perfusion and accessibility by imaging contrast agents. In addition, tumour growth is suppressed, leading to increased survival of tumor bearing mice. Importantly, the targeting of TNF α reduced the general toxicity resulting from systemic administration.

Overall, the results are interesting and convincing, the experiments have been rigorously designed. The manuscript is clearly written and well organized.

The experiments that made use of adoptive transfer of T cells into tumour bearing BALB/c nude mice are consistent with the conclusion that the targeted -TNF α induced recruitment of T cells is the mediator of the effects on ECM and associated effects (e.g. perfusion, elasticity, etc.). But it would be a stronger case if the effects of targeted TNF α administration were reversed by T cell neutralization in immunocompetent mice.

Given the positive effects of targeted TNF α administration on clinically relevant responses including penetration by imaging contrasting agents and tumor growth, the inclusion of the somewhat inconclusive and possibly under-powered elastography data in Figure 4 is unconvincing. What relevance does tumor stiffness have to these responses? Is it simply a surrogate read-out for the effects on the ECM? The data could be removed, or it should be explained more clearly why this is relevant. Does tumor stiffness *per se* actually affect vascularization, or is it more directly the inhibitory effect of the denseness of the ECM?

The effect of the targeted TNF α on tumor growth in Figure 6 is notable and convincing. Coupled with the improved perfusion by imaging reagents, it is surprising that it wasn't tested whether the targeted TNF- α would improve the therapeutic responses to chemotherapy. This would appear to be the most clinically relevant aspect of the study, and yet hasn't been tested in this system.

Minor points.

1. Figure 1A would be improved by pairing it with a visible light picture of the tissues.
2. Figure 1C isn't very well described, what does the percentage represent? Positive staining area of total area?
3. Figure 1D would be easier to appreciate if the green channel were presented separate from the red/green merge.
4. In Figure 1F, what is the statistically significant bar being compared with?
5. What are the mean elasticity values in Figure 4 for control vs treated tumors?
6. How was "survival" of mice defined in Figure 6?

1st Revision - authors' response

15 August 2019

1) In this paper the authors describe a new biotechnological tool designed to precisely deliver TNF

to the tumor ECM, trigger an inflammatory reaction and dampen the tumor growth. To reach this aim TNF is linked to the CSG peptide identified by screening a phage library on ECM (Matrigel). The experiments were mainly performed in an insulinoma transgenic model (RIP-Tag) and in a triple negative breast cancer model (4T1). In my opinion the authors' hypothesis should be validated in models, which better recapitulate the clinical need to overcome the effect of fibrosis on drug delivery, such as pancreatic ductal adenocarcinoma.

ANSWER 1:

Whilst the objective of our study is to target and destroy ECM, we have deliberately chosen models of different ECM content/intratumoral distribution to compare the efficacy of ECM degradation and anti-tumour effects.

Orthotopic 4T1 breast tumour is a clinically relevant model of 'fibroinflammation' similar to stage IV human breast cancer and it has been widely used as a triple-negative breast cancer model. In contrast, RIP1-Tag5 tumours are poorly infiltrated by immune cells (Fig 3A) and show strong non-cellular CSG binding around tumour ECM (Figs 1B and 2D and E). In addition, we have previously shown that the RIP-Tag tumours, while highly angiogenic, are poorly perfused and hardly penetrable for drugs and immune cells (Hamzah et al. Nature 2008; Hamzah et al. J Clin Invest 2008; Johansson et al. Cell Report 2016, Johansson et al. Nature Immunol 2017). Thus, the RIP-Tag5 tumour model is particularly suitable to evaluate immune infiltration, ECM depletion and changes to vessel perfusion. Indeed, the main findings of this study are the beneficial effects of ECM degradation on vessel decompression, improved perfusion and anti-tumour immune cell infiltration (Figs 4, 6 and 7).

In response to the reviewer's comment, we now include additional data generated in a transgenic model of murine HCC (ALB-Tag), which clearly indicate that TNF α -CSG promotes intratumoral T cell infiltration (new Appendix Fig S5D) and enhances lectin perfusion (new Appendix Fig S7E), much as demonstrated for 4T1 and RIP1-Tag5 tumours.

CSG binding to tumour ECM and TNF α -CSG treatment are equally effective in a murine pancreatic adenocarcinoma model (KPC mice) (R.G./J.H. unpublished; see Fig 1 [*unpublished data removed at the authors request*]). This project is part of a newly funded combination therapy study which is beyond the scope of this manuscript.

2) The data shown in Fig 1 seem to support the specificity of CSG peptide in detecting tumor ECM. The authors also demonstrate that it binds molecular determinants of basal membrane including laminin, nidogen-1 and collagen IV. The use of Matrigel to identify tumor ECM is fine because it is produced by a murine sarcoma. However this section has to be improved by a more precise identification of the molecule(s) identified by CSG.

ANSWER 2:

Our revised manuscript now includes affinity purification of matrigel proteins that bind CSG and identifies a laminin- nidogen-1 complex as CSG target (new Fig 2). In addition, we have shown in mouse and human cancers that laminin and nidogen are more abundant in tumours compared to normal tissue, and that the higher abundance in tumours correlates with CSG binding (new Fig 2; new Appendix Fig 3). Moreover, we show that the laminin-containing ECM, which in normal tissue is restricted to a thin layer of basement membrane, is not recognised by CSG (Fig 2C-E). Therefore, our revised manuscript now establishes the binding target and specificity of CSG using a spectrum of assays ranging from matrigel CSG-phage isolation, affinity chromatography, *in vitro* as well as *in vivo* binding to tumour ECM in multiple mouse models and primary human carcinoma. Hence, our data strongly support a broad applicability of CSG in cancer.

Results (page 5) and Discussion (page 10) have been extensively revised to include the new findings.

3) Furthermore the ECM of other pathological tissues besides cancer (eg lung, liver, wounded tissues) has to be tested. The demonstration of the capability of CSG to bind human cancers is just based on 4 breast cancers and therefore is weak. I suggest to extend the analysis to more patients with breast cancer, prostate cancer, pancreatic ductal adenocarcinoma and HHC. Finally, the authors have to provide a first analysis of CSG binding to normal and cancer cells.

ANSWER 3:

Please note that human tissue sample availability is limited because the CSG binding assay requires fresh patient specimens without formalin fixative. All of the samples that we have tested were collected fresh from patients within 2 hours post-surgery. In response to Reviewer's comment, we performed further studies on:

Human Breast cancer: We have tested CSG binding on new fresh samples from n=3 breast cancer patients. Revised Fig 1D and E show now analyses from a total of 7 patient samples. Our data confirm that CSG specifically binds to human breast tumours and its distribution agrees with ECM localization (new Appendix Figs S2 and S3). Importantly, normal breast tissue and normal murine organs do not bind CSG.

Human Pancreatic Adenocarcinoma (PDAC) and HCC: New Appendix Fig S2B includes quantitative CSG binding assessment on n=3 human PDAC and n=3 human HCC. These data confirm that CSG broadly recognises ECM in all mouse and human cancers tested so far. This underlines the translational potential of the CSG targeting platform for tumours, fibrotic cancers in particular, irrespective of tissue origin.

In cultured tumour cells: We assessed CSG binding *in vitro* and its receptor expression in 4T1 cells and β TC-C3H tumour cells derived from RIP-Tag mice. New Appendix Fig S3G shows that 4T1 tumour cells produced extracellular laminin and bound CSG, consistent with the CSG binding and laminin expression seen in 4T1 tumours (Appendix Fig S3E). In contrast, β TC-C3H tumour cells showed neither laminin expression nor CSG binding, indicating that the laminin structures in RIP-Tag5 tumours are derived from stroma (Figure 2D and E). Thus, CSG binding in tumours may be dictated by ECM production by neoplastic cells as well stromal cells such as CAFs.

Our result section (page 4-5) has been revised to include the new data.

We have been unable to collect fresh human prostate cancer specimens for logistical reasons, and we are currently assessing CSG binding in other fibrotic diseases including liver fibrosis. CSG binds to fibrotic liver with high abundance of ECM (see Fig 2 [unpublished data removed at the authors' request]). However, this is part of a different non-cancer related study and beyond the scope of this manuscript.

4) Fig 2. Besides the evaluation of the immune-infiltrate by immunofluorescence, a more precise evaluation of immune-profile by FACS is required. There are 3 other points: i) the type of polarization of the recruited macrophages; ii) the systemic effect of CSG-TNF: does it affect the feature of circulating or splenic lymphomononuclear cells? iii) the presence of polymorphonuclear cells in the treated tumors

ANSWER 4:

FACS quantifications of infiltrating immune cells in 4T1 tumours including T cells, macrophages (CD11b+ F480+ Ly6G-) and myeloid-derived suppressor cells (CD11b+ F480- Ly6G+) in response to TNF α -CSG, untargeted TNF α or controls were already shown in the previous version of the manuscript (these data are now in Appendix Fig S5C) and Result section (page 6). Our FACS analysis demonstrates that the increased T cell influx and generation of GrzB+ effector T cells (Fig 7D) is unique to TNF α -CSG. Macrophage infiltration is also increased in response to TNF α treatment, but as shown in Appendix Fig S5C, this occurs irrespective of CSG targeting. Thus, our assessment of tumour macrophages focused mainly on their expression of proteases (Fig 3C and Appendix Fig S6A). We also show conclusively in immunodeficient mice that T cells are major effector cells in reduction of ECM and tumour growth control.

Therefore, the main emphasis of this manuscript is on T cells. Our histology, FACS, RT-PCR, ELISA, and survival analyses consistently showed the involvement of T cells in mediating anti-tumour and ECM depletion effects associated with TNF α -CSG therapy.

Of note, TNF α -CSG therapy is highly specific to the tumour microenvironment, which is further supported by the following observations:

i) Immune cell accumulation and ECM depletion are restricted to the tumour microenvironment. Surrounding normal tissues are not inflamed, and remained intact (revised Appendix Fig S7A).
 ii) When compared to untargeted TNF α which is highly toxic, TNF α -CSG does not increase CRP levels, a marker of systemic toxicity (Appendix Fig S4E). Levels of other markers of systemic toxicity including plasma creatinine, troponin, alanine aminotransferase (ALT) and aspartate aminotransferase (AST) were also tested and are not altered by TNF α -CSG treatment (data not shown).

5) Fig 3. Besides specific staining with Abs, a general view of ECM by Masson's trichrome and Sirius red stains should be useful. Furthermore, besides the two exploited models, I think that the best one to support the working hypothesis of the authors is to test CSG-TNF on a PDAC transgenic or orthotopic model, which are used to be characterized by a frank desmoplastic reaction.

ANSWER 5:

The revised manuscript now includes new analyses (see new Appendix Fig S6B) of other ECM markers including Picosirius red, collagen I, heparan sulfate proteoglycan (perlecan), fibronectin and reticular fibroblasts (ER-TR7). Consistent with the data in the earlier version of the manuscript, all of these markers were reduced in response to TNF α -CSG treatment.

The data are now part of the revised Results section (page 7).

See also **ANSWER 1** for the choice of murine tumour models used in this study. Whilst PDAC is a highly desmoplastic cancer and already part of our follow-up CSG studies, ECM stiffness is a common clinical problem in most solid cancers making CSG a universal targeting and payload delivery platform, not only for PDAC.

6) In the same figure the authors show an inverse correlation between the expression of collagen and the CD45 recruitment. Is it just determined by a decompressing effect and an increased diapedesis of circulating CD45 cells? Or by the induction of a new and precise genetic program characterized by the appearance of chemotactic molecules? Which is the effect of CSG-TNF on the presence of myofibroblasts/CAFs?

ANSWER 6:

High ECM content and stiffness can restrict perfusion as well as prevent the migration of anti-tumour immune cells, including activated T cells, from reaching the tumour cells (cited reference Salmon et al. JCI, 2012). Thus, an inverse correlation between collagen content and immune infiltrate (now in Fig 4C and D) suggests that depletion of ECM enables deeper T cell access into tumour parenchyma. From an immunotherapeutic point of view, this is advantageous since depletion of the ECM barrier may allow better interaction between cytotoxic T cells and tumour cells as has recently been shown with anti-TGF β /checkpoint inhibitor treatment (e.g., Mariathasan et al. Nature 2018).

Vessel decompression may improve systemic access in tumours and our data (Fig 6E and F) consistently showed increased tumour perfusion and enhanced systemic uptake of theranostic agents.

We have not evaluated the effect of TNF α -CSG on CAFs other than reduction in the reticular fibroblast marker, ER-TR7 in TNF α -CSG treated tumours (new Appendix Fig S6B).

7) Fig 2 -4. Are the phenotypes here described reverted along the time and after the treatment interruption?

ANSWER 7:

Our data indicate that a short treatment with 5 consecutive daily i.v. injections of TNF α -CSG achieved intratumoral effects (Appendix Fig S8A and B; Appendix Fig S9A) and sustained survival benefits in RIP-Tag5 mice (up to 9 weeks; Fig 7A) and delayed growth of 4T1 tumour (up to two weeks; Fig 7B) after treatment.

8) Fig 4 clearly shows that CSG-TNF modify the tumor stiffness. Is there any correlation between this effect and the proliferative and apoptotic index of cancer cells? Can the authors mimic the in

vivo results by analyzing the *in vitro* behavior of cancer cells maintained on matrixes with different stiffness?

ANSWER 8:

The microelastography technique allows us to measure structural heterogeneity of tumour stiffness (now in Fig. 5). Data in Fig 5C (and new Appendix Table S1) indicate the mean stiffness is unchanged relative to reduced heterogeneity, in response to TNF α -CSG. However, the “stiffest” and “softest” regions are “normalized” in TNF α -CSG-treated tumours. The stiffest regions correlate with ECM content (shown for 4T1 in Fig. 5E and RIP1-Tag5 tumours in new Appendix Fig S7B). This is in agreement with our previous study (Kennedy et al. *Cancer Res* 2015; 75(16), 3236-45) and of others (Plodinec et al. *Nature Nanotech*, 2012; 7, 757-65) which show tumour stiffness strongly correlates with ECM/stromal density and is not associated with tumour cell density. A recently published study by Riegler et al. (*Clin Cancer Res* 2019) also show that stiffness of 4T1 tumours (measured by ultrasound elastography) has no correlation with apoptotic regions in the centre of 4T1 tumour.

Studying the effects of TNF α -CSG on stiffness of artificial matrices *in vitro* could be interesting but is outside the scope of this already quite extensive study.

9) Fig 7. Do CD8 cells isolated from treated tumors exert an *in vitro* cytotoxic effect on cancer cells

ANSWER 9:

We prioritized our analyses to more biologically relevant *in vivo* studies and did not examine cytotoxic effects of tumour infiltrating CD8+ T cells *in vitro*. The evidence of cytotoxic CD8 T cell involvement in TNF α -CSG treated tumours came from extensive *in vivo* studies, specifically:

- i) A 9-fold increase in CD8+ T cells expressing the cytotoxic marker (granzymeB) in response to TNF α -CSG (Fig 7D).
- ii) TNF α -CSG treatment resulted in enhanced survival and tumour shrinkage (Fig 7). It also reduced tumour cell proliferation and increased tumour cell apoptosis *in vivo* at doses that did not reduce tumour cell viability *in vitro*, (Appendix Fig S8).
- iii) The intratumoral and treatment effects were attenuated in T cell-deficient mice and adoptive transfer of T cells rescued all therapeutic effects (Fig 8 and Appendix Fig S8E).

Reviewer 2:

The manuscript by Yeow et al. describes a potentially highly interesting approach, based on a tumour ECM-homing CSG peptide coupled to the cytokine, tumour necrosis factor alpha, to induce immune cell infiltration and ECM degradation. The consequences of desmoplastic tumor microenvironment for tumor aggressive properties and therapy resistance are being increasingly appreciated. Therefore, by providing means to dissolve the desmoplastic ECM, the reported approach could have valuable clinical implications for the treatment of different cancers. However, there are some concerns and questions which should be addressed in order to strengthen the manuscript.

Specific comments:

10) A major structural ECM component in the desmoplastic tumour ECM is collagen I, in addition to other fibrillar collagens, fibronectin etc, which co-localization with FAM-CSG, and degradation/loss after the treatments should be carefully addressed. This is a critical point to understand the biological and possible clinical relevance of the developed tumour targeting approach: Therefore, in human and mouse tumors, FAM-CSG binding/homing should be compared to interstitial collagen and fibrillar ECM structures of the desmoplastic tumor microenvironment rather than the basement membrane components only (in addition to immunohistochemistry e.g. Trichrome stain could be used).

ANSWER 10:

New data: Our revised manuscript now includes the identification of laminin-nidogen-1 complexes as the binding target for CSG and their accumulation/localization in tumour ECM (new Fig 2 & Appendix Fig 3; see ANSWER 2). Our new data clearly show that the overexpressed laminin-nidogen-1 complexes in tumours are not restricted to the normal localization of basement

membranes (no overlaps with CD31 staining; Fig 2C) but include ECM components positive for collagen I and fibrillary collagens (trichrome staining) (Appendix Fig S3B and C). Importantly, we show these avascular ECM complexes are the main target recognised by CSG (Fig 2 D-E; Appendix Fig S3D -F), which is consistent across mouse and human tumours.

Our new data have now addressed Reviewer's request, specifically we show:

- i) Colocalization of CSG with collagen I in RIP1-Tag5 tumours (Appendix Fig S3F).
- ii) Colocalization of laminin with collagen I in RIP1-Tag5 and 4T1 tumours (Appendix Fig S3B).
- iii) Colocalization of laminin with collagen I in human breast tumour, and serial-staining of tumour tissues indicating laminin and fibrillary collagens (trichrome staining) are part of the same ECM structure; Appendix Fig S3C).

These data show that CSG is a specific molecular agent to target desmoplasia, and this is consistent in multiple mouse and human tumours.

11) Figure 1B: What are the FAM-CSG+ round structures in 4T1 T; do they represent basement membranes, fibrillar ECM or something else? Are the FAM-CSG+ areas in RT5 T acellular? More detailed histological images and result description will allow better consideration of the specificity of CSG homing. Co-localization should be compared between different types of ECM components such as the BM proteins, fibrillar collagens and fibronectin. Double staining with markers for cancer-associated fibroblasts and/or tumor cells would also help in defining more specifically the targeted structures in tumors/tumor-associated stroma.

ANSWER 11:

In response to Reviewer's request, a comprehensive colocalization analysis of CSG binding to laminin, nidogen-1 and collagen-IV in 4T1 tumours is now shown in new Appendix Fig S3E. Colocalization with collagen I is shown in Appendix Fig S3B, as described in ANSWER 10. In addition, we also show extracellular CSG binding on cultured 4T1 cells that produced laminin (new Appendix Fig S3G).

In RIP1-Tag5 tumours, CSG and TNF α -CSG recognised stromal-associated ECM that is predominantly non-cellular (area negative for nuclei staining, Fig 1B and Appendix Figs S4C and S3G); this is one of the reasons why RIP1-Tag5 is an excellent model to demonstrate the effect of TNF α -CSG on ECM depletion.

We were unable to find a reliable marker to identify cancer-associated fibroblasts in RIP1-Tag5 insulinoma, 4T1 tumours as well as in human tumours; the tissues were poorly stained for anti-fibroblast activation protein (anti-FAP). Our analysis of mouse tumours using antibody against ER-TR7 (a marker for reticular fibroblasts and fibres), shows some colocalization with CSG mainly in non-cellular ECM areas (new Appendix Fig S3F).

12) The authors state that CSG homing is specific for ECM (e.g. in the following citation from Abstract): "This peptide specifically targets tumour ECM of mouse and human carcinomas and selectively delivers a recombinant TNF α -CSG fusion protein to tumour ECM in tumour-bearing mice." However, cellular localization patterns are also shown in e.g. Fig S1G. How was cellular homing and delivery addressed/ruled out? Could cell dissociation followed by FACS be used to show the binding or lack of binding/delivery to cells?

ANSWER 12:

As indicated in ANSWER 11, the cells that express laminin, such as 4T1 cells, may show some binding. As the newly synthesized ECM components are closely associated with the cells that make them, it is difficult to determine where the cell surface ends and the ECM begins. As we show that CSG binds to laminin-nidogen complexes in vitro and in areas free of tumor cells in vivo, there is no reason to assume any cell binding or effect that would be independent of the laminin-nidogen binding. Moreover, direct exposure of cultured 4T1 cells to TNF α -CSG, even at concentrations 7 to 18-fold higher than the estimated in vivo levels, did not alter protease expression (except to some extent that of MMP3; Appendix Fig S6A) or reduce cell viability (Appendix Fig S8C), suggesting that the TNF α -CSG effect is not mediated by direct action on tumour cells.

The abstract has been reworded:

"This peptide binds to laminin-nidogen complexes in the extracellular matrix (ECM) of mouse and human carcinomas with little or no peptide detected in normal tissues, and it selectively delivers a recombinant TNF α -CSG fusion protein to tumour ECM in tumour-bearing mice."

13) Figure 1E shows a pattern of specific CSG binding in one human breast tumor section. Will similar ECM-patterns be seen in majority of human breast cancers and other types of human cancer?

ANSWER 13:

We have assessed CSG binding in 7 breast cancer specimens and 3 samples of human HCC and PDAC. See also **ANSWER 3**.

14) Figure 3: The loss/degradation of proteins like collagen I/III/V and fibronectin which are major components of the desmoplastic tumour ECM should be compared and quantified in addition to the basement membrane proteins analyzed. Collagen IV is a network-forming collagen and a component of basement membranes; therefore it is unclear what is the relevance of reduced Col-IV (or the other BM proteins), used as the main read-out throughout the manuscript, for tumour stiffness and physical properties.

ANSWER 14:

As indicated in **ANSWER 5**, the revised manuscript now includes new analyses (**Appendix Fig S6B**) of ECM markers including Picosirius red, collagen I, fibronectin, and reticular fibroblasts (ER-TR7), in addition to another basement membrane component, the heparan sulfate proteoglycan or perlecan. Consistent with the data in the earlier manuscript, all of these markers were reduced in response to TNF α -CSG treatment.

15) Does the degradation of (vascular) basement membranes affect vascular leakage or the amount (area) or proliferation of the CD31+ vascular cells?

ANSWER 15:

Our data show ECM degradation enhances the *functionality* of CD31+ tumour blood vessels due to decompression and vessel dilation. As illustrated in **Fig 6B**, lectin uptake in TNF α -CSG treated tumours showed significant overlap with CD31 positive blood vessels, in comparison to the control tumours that show CD31+ vessels that were poorly perfused with lectin. There was no difference in CD31+ vessel numbers in control and TNF α -CSG –treated tumours, suggesting that TNF α -CSG has no effect on vessel proliferation (**new Appendix Fig S7D**).

The TNF α -CSG treated tumours show widespread distribution of nanoparticles (\approx 30 nm, indication that decompression of tumour vessels also increases vessel leakiness (**Fig 6F**).

16) The authors mention in discussion that CSG contains a potential CendR motif for transcellular and transtissue transport pathway. It would be important to address, if the presented peptide targeting/homing pattern is due to the specific ECM binding capability of CSG or if it involves the transcellular and transtissue transport pathway.

ANSWER 16:

The focus of our paper is the CSG specificity to tumour ECM and delivery of TNF α into tumours and the ensuing anti-tumour effects. We will evaluate the peptide penetration mechanism in future studies. In our discussion, we merely wanted to share the information about a potential CendR motif with researchers who may want to use the CSG peptide for full disclosure.

17) All the presented results and discussion considered, it is difficult to draw concise conclusions about the various aspects of the reported phenotype with regards to the underlying molecular mechanisms or the most relevant clinical implications. For example, is the "therapeutic benefit" of TNF α -CSG treatment dependent on the altered ECM (or vascular) phenotype, or on the immune cell influx only? Were the immune cell influx, decreased tumour growth and proliferation associated with increased tumor cell killing/death?

ANSWER 17:

TNF α -CSG treatment gives solid tumours a one-two punch, immune cell entry and ECM degradation within the tumour. Thus, immune cell infiltration and ECM degradation are not separate

entities. The effects of TNF α -CSG can be used in two ways:

- i) Pre-treatment with TNF α -CSG may sensitize inaccessible tumours deeply for better detection. Here, we demonstrate the ECM degradation improves perfusion exemplified by the nanoparticle contrast agent.
- ii) We envision the use of TNF α -CSG to convert an immunologically “cold” tumour without adequate immune infiltration for an immune response into a “hot” one capable of triggering such a response, particularly in combination with check-point inhibitor antibodies. The effect of TNF α -CSG on the ECM may make it easier for such antibodies, or other therapeutic agents to gain access to a tumour, particularly in desmoplastic cancers which have high ECM content and compressed vessels.

We have revised the discussion to point out these potential benefits more clearly (page 13).

18) The importance of the ECM degradation for the described TNF α -CSG treatment outcome could be investigated by testing the effects of relevant protease inhibitors on tumor growth, immune cell influx and vascular perfusion. Or if the vascular phenotype including increased perfusion suggestive of conditions for improved drug delivery was a key aspect of the TNF α -CSG treatment-induced phenotype, a proof-of-principle combination treatment with relevant anti-cancer drugs would strengthen the study.

ANSWER 18:

The main strength of our therapeutic approach, which was never been described previously, is that TNF α -CSG triggers secretion of a protease cocktail locally within the tumour. Currently, systemic therapy with protease cocktails is not a viable option because of very significant systemic toxicity. Thus, our approach provides a safe alternative to degrade tumour ECM effectively. It is unlikely that one specific family of proteases is responsible for the complex biological effects that we observed in vivo. It is therefore difficult to rationalise which protease inhibitor/s to use to counteract the ECM depletion effects.

In this manuscript, our main focus was to demonstrate the use of TNF α -CSG as an immunotherapeutic (Figs 7 and 8) and also, as an agent to sensitize inaccessible tumours for improved detection through the delivery of imaging agents (Fig 6). There have been a large number of studies that show the effectiveness of vessel normalization therapies (which also improve tumour perfusion) in combination with chemotherapy (see review Jain, Cancer Cell 2014). The vessel decompression effect achieved by TNF α -CSG is different to all vessel normalization strategies published so far and the applications in this manuscript were deliberately designed as a point of differentiation to vessel normalization therapy. Nevertheless, we are keen to pursue follow up drug combination therapies which we envision to conduct in PDX models.

19) Minor points: Which of the several laminin(s) were analyzed? Figure 1E: Imaging/image processing; the background in Hu N FAM-CSG and FAM-ARA appear darker than in the other samples.

ANSWER 19:

Laminin consists of three polypeptide chains, each of which comes in different subtypes. The elution of the affinity matrix (i.e. extract of basement membrane from EHC mouse tumour) with soluble CSG peptide indicate 2 laminin bands, which were identified by mass-spectrometry as laminin subunits alpha-1 and gamma-1 (new Fig 2A), in addition to weaker but traceable band corresponding to nidogen-1. For all histology analysis of mouse and human laminin, we use a polyclonal antibody raised against laminin extracted and purified from EHC mouse tumour tissues (Anti-Laminin Antibody, Merck Millipore AB2034). The polyclonal antibody is likely to detect a number of different laminins in addition to laminin alpha-1 chain.

Figure 1E: We have reproduced the peptide binding studies Fig 1D and E and Appendix Fig 2 in which nuclei were stained with methyl green which provides better contrast.

Reviewer 3:

In this manuscript, the authors describe how a targeted form of TNF α can increase the infiltration of tumor-associated extracellular matrix (ECM) by immune cells. The targeting sequence

was identified by screening phage-displayed peptide libraries with an *in vivo* approach that was based on selective tumor recognition. As a consequence, the ECM becomes degraded and less stiff, allowing for greater tumor perfusion and accessibility by imaging contrast agents. In addition, tumour growth is suppressed, leading to increased survival of tumor bearing mice. Importantly, the targeting of TNF α reduced the general toxicity resulting from systemic administration. Overall, the results are interesting and convincing, the experiments have been rigorously designed. The manuscript is clearly written and well organized.

20) The experiments that made use of adoptive transfer of T cells into tumour bearing BALB/c nude mice are consistent with the conclusion that the targeted -TNF α induced recruitment of T cells is the mediator of the effects on ECM and associated effects (e.g. perfusion, elasticity, etc.). But it would be a stronger case if the effects of targeted TNF α administration were reversed by T cell neutralization in immunocompetent mice.

ANSWER 20: Our studies performed in mice completely devoid of T cells (now in Fig 8) show that T cells are required for the anti-tumour effects. Whilst T cell neutralization would indeed be a different experimental approach, the outcome would still be the same and therefore does not ethically justify the use of another cohort of mice.

21) Given the positive effects of targeted TNF α administration on clinically relevant responses including penetration by imaging contrasting agents and tumor growth, the inclusion of the somewhat inconclusive and possibly under-powered elastography data in Figure 4 is unconvincing. What relevance does tumor stiffness have to these responses? Is it simply a surrogate read-out for the effects on the ECM? The data could be removed, or it should be explained more clearly why this is relevant. Does tumor stiffness *per se* actually affect vascularization, or is it more directly the inhibitory effect of the denseness of the ECM?

ANSWER 21:

We assessed changes to tumour stiffness in response to TNF-CSG in this study because:

- i) Increased stiffness reflects the pathological state of malignant tissue that influences cell behaviour in tumour (see review by Lampi & Reinhart-King; *Sci. Trans. Med.* 2018, 10, 1-14).
- ii) Relationship between increased tumour stiffness, reduced tumour perfusion and reduced drug delivery is well-documented (References: Provenzano et al. *Cancer Cell* 2008, 21:418-29; Netti et al *Cancer Res.* 2000, 60:2497–503; Riegler et al. *Clin Cancer Res* 2019).

In this study, we consistently show TNF α -CSG treatment reverses ECM fibrosis, tumour stiffness, perfusion and drug access.

We agree with the Reviewer that reduction in tumour stiffness may relate to the overall anti-tumour effect of TNF α -CSG on tumour cells. However, we (Kennedy et al. *Cancer Res* 2015: 75(16), 3236-45) and others (Plodinec et al. *Nature Nanotech*, 2012: 7, 757-65; Riegler et al. *Clin Cancer Res* 2019) have shown previously that tumour stiffness, imaged by microelastography, ultrasound-based elastography or atomic force microscopy, correlates strongly with ECM/stromal density, rather than with tumour cells. In particular, breast cancer cells are typically softer than normal breast cells and thus are captured as least stiff regions, even when histologically presented as densely packed cell clusters. In this study, there is no difference in the mean tumour stiffness between control and TNF α -CSG treated tumours. However, we see a clear reduction of the stiffest regions that correspond to ECM. Thus, in 4T1 and RIP1-Tag5 tumours, reduction of stiffest regions correlates with decreased in ECM content (Fig 5 and new Appendix Fig S7B).

We have revised the Results section (page 8) to point on stiffness data more clearly.

22) The effect of the targeted TNF α on tumor growth in Figure 6 is notable and convincing. Coupled with the improved perfusion by imaging reagents, it is surprising that it wasn't tested whether the targeted TNF- α would improve the therapeutic responses to chemotherapy. This would appear to be the most clinically relevant aspect of the study, and yet hasn't been tested in this system.

ANSWER 22:

Given the topicality of tumour immune therapy, we think the effects on immune cell recruitment are the most important aspect of the study (see **ANSWER 18**). Moreover, this manuscript

comprehensively covers (i) the discovery of ECM targeting peptide and its binding properties to a spectrum of mouse and human tumours, (ii) identification of its binding target, and (iii) a proof-of-principle use to deliver a drug with inherent therapeutic value via immune activation and (iv) local ECM depletion. Follow up drug combination therapies will be the subject of new studies using PDX models.

23) Minor points. 1. Figure 1A would be improved by pairing it with a visible light picture of the tissues. 2. Figure 1C isn't very well described, what does the percentage represent? Positive staining area of total area? 3. Figure 1D would be easier to appreciate if the green channel were presented separate from the red/green merge. 4. In Figure 1F, what is the statistically significant bar being compared with? 5. What are the mean elasticity values in Figure 4 for control vs treated tumors? 6. How was "survival" of mice defined in Figure 6?

ANSWER 23:

1. Fig 1A: Revised Fig 1A now includes a photograph of the corresponding tissues under visible light.

2. Fig 1C: The figure legend has been corrected and changes are underlined:
Figure 1. CSG specifically recognises mouse and human tumours, and binds to ECM. (A-C) Mice bearing orthotopically implanted 4T1 breast cancers and RIP1-Tag5 (RT5) tumours were i.v. injected with 0.1 μmol of FAM-CSG, and tissues were collected after 1 h circulation. (A) Photographic image of tissues from 4T1 tumour-bearing mouse under bright light and UV-illuminator. (B) and (C) Distribution of FAM-CSG in different tissues including tumours (4T1 T and RT5 T), kidney (K), vertebrae (V), lung (LG), liver (LV), intestine (I), muscle (MU), spleen (SP), heart (H), pancreas (P), brain (B), lymph node (LN) and skin (SK), detected by immunoperoxidase staining with anti-FITC antibody. Representative staining (brown) is shown for each tissue in B and as mean \pm SEM of percentage area per tissue section stained with anti-FITC antibody (n=3; *P<0.05, ** P<0.005, tumour compared to other tissues except kidney by one-way ANOVA test with Tukey correction) in C. Scale bar: 100 μm . (D and E) Human tumour and normal breast tissues: 8 μm serial tissue sections were incubated for 20 min with 1 μM FAM-CSG or FAM-ARA, in the presence or absence of 1 mM unlabeled CSG peptide. CSG (brown) was detected as in panel B. (D) Representative micrographs of corresponding tissues stained with anti-FITC antibody (brown) are shown for an individual patient sample. (E) Bar charts show mean \pm SEM of percentage area per tissue section stained with anti-FITC antibody (n=4 Hu NB and n=7 Hu BT; ***P<0.001 and ****P<0.0001 by one-way ANOVA test with Tukey correction).

3. Fig 1D (co-staining analysis of CSG and ECM markers) is no longer part of revised Fig 1. New Fig 2 and Appendix Fig S3 describe the receptor identification and compare the ECM targets in their abundance and distribution relative to other ECM markers and tumour blood vessels (see ANSWER 2).

4. Statistically significant bar in Fig 1F (now in Fig 1E): See revised figure legend for Fig 1E above (underlined).

5. Mean elasticity values in Figure 4 for control vs treated tumors?

Stiffness/elasticity values (Kpa) for each 4T1 and RIP1-Tag5 tumours measured by OCT-microelastography and mean \pm SE are now shown in Appendix Table 1. We have revised the Results section (page 8) to point on stiffness data more clearly.

6. Definition of survival of mice in Fig 6 (now in Fig 7):

Survival of 4T1 tumour-bearing mice was based on the external measurement of tumour volume; survival endpoint is defined as a tumour size of 1,000 mm^3 which is our ethical limit as defined by the Animal Ethics Committee of the University of Western Australia. Please note that most studies in the US/Asia and Europe can be continued until tumour growth has reached a volume up to 3,000 mm^3 .

Survival of RIP1-Tag5 mice was based on welfare impact scores as the mice succumb to insulinoma between 26 to 32 weeks of age due to hypoglycemia. We administered $\text{TNF}\alpha$ -CSG treatment at 25 weeks of age, at which time tumour development was advanced.

Thank you for the submission of your revised manuscript to EMBO Molecular Medicine. We have now received the enclosed reports from the referees that were asked to re-assess it. As you will see the reviewers are now globally supportive and I am pleased to inform you that we will be able to accept your manuscript pending the following final amendments:

1) Please address the comments of referees 1 and 2, in writing. At this stage, we'd like you to discuss referee's 1 points and if you do have data at hand, we'd be happy for you to include it, however we will not ask you to provide any additional experiments at this stage. Referee 2's comments must be discussed within the article as well.

Please submit your revised manuscript within two weeks. I look forward to seeing a revised form of your manuscript as soon as possible.

***** Reviewer's comments *****

Referee #1 (Comments on Novelty/Model System for Author):

In the first revision round I clearly state the best model is PDAC. The authors just show a picture suggesting the feasibility of this request.

Referee #1 (Remarks for Author):

The MS has been greatly improved. The marginal effort done by the authors to understand the role of CAF in my opinion represents a limit for a paper focused on ECM targeting. Furthermore I disagree with the opinion that PDAC will be "part of our follow-up CSG studies" as stated by the authors. On translational point of view a PDAC model should be the first model to be analyzed, in particular for the features and the aims of the Journal.

Referee #2 (Remarks for Author):

The revised manuscript by Yeow et al. has improved substantially. New important data has been added identifying laminin and nidogen as the binding targets for the CSG peptide. The co-localization data in this revised manuscript collectively support the CSG homing and binding to the corresponding ECM protein structures in mouse and human tumors. New data has also been added to support the broad depletion of the tumor ECM including Col-I, perlecan, and fibronectin, after the TNF-CSG treatment. Therefore, these results describe a new interesting mechanism for targeting desmoplastic tumors.

Some of the raised questions and concerns remain, however, and these would seem important to address by more critical statements and discussion in the manuscript.

1. In the models analyzed, the laminin accumulation and consequent CSG homing seem to most convincingly occur at irregular perivascular ECM areas (clear in Fig. 2D), and these areas seem broader compared to homing of previously described CREKA peptide (Fig. 2E). The ECM depletion acquired by TNF-CSG is also clearly efficient, as compared to similar approach by RGR-based targeting (Fig. 4A; could it be clarified in the text why one peptide is used to compare homing and the other ECM depletion?). In RIP1-Tag5 model this TNF-CSG treatment further leads to robust immune cell infiltration, proteolysis and ECM depletion responses, including the loss of Col-I (Fig. S6B).

Therefore, as the TNF-CSG treatment was not tested, and the FAM-CSG colocalization with Col-I was not assessed in really Col I-rich desmoplastic tumors, like the one (human breast cancer) shown in Fig. S3C, it seems unnecessary to try convincing that this peptide will home ECM throughout the desmoplastic tumors. Should be better justified and likely enough to conclude that homing to tumor

laminin triggers immune cell infiltration into the tumors and protease induction, which leads to ECM depletion, including the loss of laminin, collagen, fibronectin etc.

2. The presented strong conclusions of increased vascular functionality, i.e. vessel dilation and perfusion coupled with increased leakiness seem to this reviewer also as a strong conclusion without rigorous analysis of the endothelium and vessel wall alterations. Increased width of lectin positive structures and lectin-CD31 overlap after the treatment could be related to the degradation/depletion of the subendothelial basement membranes, and thus altered vessel wall integrity, endothelial cell morphology, sprouting and/or number/proliferation. Such changes may be difficult to detect by simply quantifying CD31 positive vessels after relatively short treatments.

3. The new data in Fig. S6B indicates that ECM depletion was accompanied by loss of ER-TR7 positive fibroblasts. This result and the possible impact of this to the heterogeneous tumor stiffness and other alterations in the treated tumors would seem worth critical consideration/discussion.

Referee #3 (Remarks for Author):

I am satisfied with the revisions made to this manuscript, and recommend acceptance.

2nd Revision - authors' response

26 September 2019

Reviewer 1

In the first revision round I clearly state the best model is PDAC. The authors just show a picture suggesting the feasibility of this request.

The MS has been greatly improved. The marginal effort done by the authors to understand the role of CAF in my opinion represents a limit for a paper focused on ECM targeting. Furthermore I disagree with the opinion that PDAC will be "part of our follow-up CSG studies" as stated by the authors. On translational point of view a PDAC model should be the first model to be analyzed, in particular for the features and the aims of the Journal.

ANSWER 1:

Please note that our manuscript was revised following the Editor's specific instruction to "*make the paper stronger by addressing all issues pertaining to the clinical and translational effects*". Thus, our revised paper focussed on CSG binding to its ECM target in human cancers including breast, liver and PDAC. Furthermore, the improvement in immune infiltration and vascular perfusion in response to TNFa-CSG were also shown in a transgenic mouse model of HCC, in addition to existing data in orthotopic model of 4T1 breast tumour as well as transgenic model of RIP1-Tag5 insulinoma. We clearly indicated the relevance of RIP1-Tag5 tumour model to demonstrate the effect of TNFa-CSG on depletion of non-cellular ECM components.

Marginal effort done by the authors to understand the role of CAF:

Our manuscript was not about understanding the role of CAFs in cancer but focused on development of molecular agents to specifically target tumour ECM for immunotherapeutic and ECM depletion strategies. We showed that our drug, TNFa-CSG, affects complexed ECM structures in advanced tumours. Nevertheless, we acknowledge in the revised discussion (see also **ANSWER 5**) the potential effects of TNFa-CSG on fibroblasts which were not analysed in this study.

A PDAC model should be the first model to be analyzed:

While we share the same enthusiasm as **Reviewer 1** to assess the effects of TNFa-CSG in PDAC models (which is part of our ongoing therapeutic studies), there is no justification that a PDAC model would be a prerequisite for analysing the effects of our agents. High ECM content is a common feature of many solid tumours and not exclusive to PDAC. As shown in this study and others (e.g. Kirtane et al 2017 and Rahbari et al. 2016, cited in this manuscript), ECM in breast, liver, skin, lung and metastatic colorectal cancers, can be targeted for therapeutic intervention.

Reviewer 2

The revised manuscript by Yeow et al. has improved substantially. New important data has been added identifying laminin and nidogen as the binding targets for the CSG peptide. The colocalization data in this revised manuscript collectively support the CSG homing and binding to the corresponding ECM protein structures in mouse and human tumors. New data has also been added to support the broad depletion of the tumor ECM including Col-I, perlecan, and fibronectin, after the TNF-CSG treatment. Therefore, these results describe a new interesting mechanism for targeting desmoplastic tumors.

Some of the raised questions and concerns remain, however, and these would seem important to address by more critical statements and discussion in the manuscript.

1. In the models analyzed, the laminin accumulation and consequent CSG homing seem to most convincingly occur at irregular perivascular ECM areas (clear in Fig. 2D), and these areas seem broader compared to homing of previously described CREKA peptide (Fig. 2E). The ECM depletion acquired by TNF-CSG is also clearly efficient, as compared to similar approach by RGR-based targeting (Fig. 4A; could it be clarified in the text why one peptide is used to compare homing and the other ECM depletion?). In RIP1-Tag5 model this TNF-CSG treatment further leads to robust immune cell infiltration, proteolysis and ECM depletion responses, including the loss of Col-I (Fig. S6B).

ANSWER 2:

CREKA peptide is used as control to compare homing, and RGR-based targeting (TNF α -RGR) to compare ECM-depletion:

Please note that RGR homing to tumour blood vessels has been extensively described in our previous publication (Hamzah et al JCI 2009). In this manuscript, CREKA which also binds to tumour blood vessels was used as a control in the homing study, to minimise duplication of already published data on the RGR peptide.

However, it was important to compare the respective TNF α fusion compounds, namely TNF α -CSG, to our previously established TNF α -RGR (Johansson et al. PNAS 2012), to demonstrate that ECM depletion was exclusively achieved by TNF α -CSG. Please note, the treatment doses and frequencies of TNF α -CSG and TNF α -RGR administered in this study (2 or 5 ug/injection/day, consecutive daily \times 5) were different from those of TNF α -RGR published in Johansson et al. PNAS 2012 (2 ug/injection, twice/week for up to 10 weeks).

In response to **Reviewer**'s request, we have revised the following sentence in page 6:

“In contrast, our previously described TNF α fused to the vessel targeting peptide RGR (Hamzah et al., 2008; Johansson et al., 2012) at similar total doses, caused immune cell infiltration around tumour blood vessels (Appendix Fig S5B).”

Therefore, as the TNF-CSG treatment was not tested, and the FAM-CSG colocalization with Col-I was not assessed in really Col I-rich desmoplastic tumors, like the one (human breast cancer) shown in Fig. S3C, it seems unnecessary to try convincing that this peptide will home ECM throughout the desmoplastic tumors. Should be better justified and likely enough to conclude that homing to tumor laminin triggers immune cell infiltration into the tumors and protease induction, which leads to ECM depletion, including the loss of laminin, collagen, fibronectin etc.

ANSWER 3:

Modified conclusion on homing to tumour laminin:

We appreciate Reviewer's comment and suggestion to modify this particular conclusion. We have revised the manuscript text to incorporate Reviewer's suggestion (underlined).

Results (page 9): “In summary, our findings show that treatment of tumour-bearing mice with TNF α targeted to tumour laminin-nidogen complexes is well tolerated and induces tumour infiltration of immune cells, which in turn results in loss of tumour ECM, improved tumour perfusion, reduced tumour burden, and enhanced overall survival”.

Discussion (page 10): “We have constructed a recombinant TNF α fusion protein that specifically localises to tumour laminin-nidogen complexes when injected systemically into mice bearing desmoplastic tumours.”

2. The presented strong conclusions of increased vascular functionality, i.e. vessel dilation and perfusion coupled with increased leakiness seem to this reviewer also as a strong conclusion without rigorous analysis of the endothelium and vessel wall alterations. Increased width of lectin positive structures and lectin-CD31 overlap after the treatment could be related to the degradation/depletion of the subendothelial basement membranes, and thus altered vessel wall integrity, endothelial cell morphology, sprouting and/or number/proliferation. Such changes may be difficult to detect by simply quantifying CD31 positive vessels after relatively short treatments.

ANSWER 4:

Improving vascular functionality was only stated in **Rebuttal 1 ANSWER 15** in response to the Reviewer’s question. Whereas, in our manuscript we specifically refer to the TNFa-CSG effect as improving tumour perfusion, for instance:

- (i) increased vessel dilation based on quantification of lectin-painted vessels (Fig 6A and B),
- (ii) in vivo real-time gadolinium uptake in tumours (Fig 6D), and
- (iii) in vivo uptake of IO NPs with widespread particle distribution in tumour parenchyma and not restricted to tumour blood vessels (Fig 6E and F).

However, the Reviewer is correct, these effects are likely short-lived:

- i) ECM/basement membrane depletion may compromise vessel wall integrity, endothelial cell morphology, proliferation and sprouting.
- ii) Importantly, since TNFa-CSG triggers immune-mediated anti-tumour effects, the therapy is not meant to maintain intact tumour vascularity long term.

Nonetheless, the main conclusion of our manuscript is that the effects of TNFa-CSG on anti-tumour immune cell infiltration and ECM degradation are not separate entities. Improved vessel perfusion following TNFa-CSG treatment may allow improved efficacy in drug combination therapy (**Discussion, page 13**).

3. The new data in Fig. S6B indicates that ECM depletion was accompanied by loss of ER-TR7 positive fibroblasts. This result and the possible impact of this to the heterogenous tumor stiffness and other alterations in the treated tumors would seem worth critical consideration/discussion.

ANSWER 5:

ER-TR7 staining and its colocalization with CSG binding were mostly observed in the basement membrane areas associated with blood vessels (Fig EV2F). Therefore, it is unlikely that reduced ER-TR7 signals in TNFa-CSG-treated tumours (Fig EV4B) are related to overall altered tumour stiffness throughout the tissue. In this manuscript, we did not evaluate the effect of TNFa-CSG on fibroblasts, and we acknowledged that TNFa-CSG may also affect fibroblasts.

In response to Reviewer’s request, we have included the following revised (underlined) text in the **Discussion (page 11)**.

“Systemic delivery of ECM-degrading enzymes such as PEGPH20 (pegylated hyaluronidase) has been used to degrade tumour ECM components for the purpose of enhancing perfusion and access to solid tumours (Caruana et al., 2015; Guedan et al., 2010; Kirtane et al., 2017; Provenzano et al., 2012; Rahbari et al., 2016). The TNFa-CSG strategy differs from these approaches in that the ECM reduction following TNF α -CSG therapy is a result of immune cell infiltration and local production of a cocktail of ECM-degrading proteases by these immune cells. The ECM depletion effects include the loss of laminin, nidogen-1, collagen IV, collagen I, fibrillar collagen, fibronectin, perlecan and potentially fibroblasts which are positive for ER-TR7. Whilst our data clearly indicate that TNFa-CSG can remove ECM components in advanced tumours, one limitation of this study is that the potential suppressive effect by TNFa-CSG on tumour fibroblasts and their secretion of ECM components was not addressed.”

Reviewer 3

I am satisfied with the revisions made to this manuscript, and recommend acceptance.

Corresponding Author Name: Juliana Hamzah
 Journal Submitted to: EMBO MOL MED
 Manuscript Number: EMM-2019-10923-V2